# SimpleGPT: Improving GPT via A Simple Normalization Strategy

Marco Chen [* 1]  Xianbiao Qi [* † 2]  Yelin He [2]  Jiaquan Ye [2]  Rong Xiao [† 2]

## Abstract

In this work, we revisit Transformer optimization through the lens of second-order geometry and establish a direct connection between architectural design, activation scale, the Hessian matrix, and the maximum tolerable learning rate. We introduce a simple normalization strategy, termed SimpleNorm, which stabilizes intermediate activation scales by construction. Then, by analyzing the Hessian of the loss with respect to network activations, we theoretically show that SimpleNorm significantly reduces the spectral norm of the Hessian, thereby permitting larger stable learning rates. We validate our theoretical findings through extensive experiments on large GPT models at parameter scales 1B, 1.4B, 7B and 8B. Empirically, SimpleGPT, our SimpleNorm-based network, tolerates learning rates $3\times$-$10\times$ larger than standard convention, consistently demonstrates strong optimization stability, and achieves substantially better performance than well-established baselines. Specifically, when training 7B-scale models for 60K steps, SimpleGPT reduces the training loss from 2.290 to 2.208 compared with Llama2 with QKNorm. Our code is available at https://github.com/Ocram7/SimpleGPT.

## 1. Introduction

Transformer-based large language models (LLMs) (Radford et al., 2018; 2019; Brown et al., 2020; Touvron et al., 2023a;b; Dubey et al., 2024; Chowdhery et al., 2023; Liu et al., 2024; Team, 2023) have achieved state-of-the-art performance across a wide range of tasks. As these models scale in depth and width, optimization stability increasingly constrains performance and scalability. Many architectural components in modern Transformers—such as residual con-

nections (He et al., 2016), normalization layers (Ioffe & Szegedy, 2015; Ba et al., 2016; Zhang & Sennrich, 2019; Large et al., 2024), and nonlinear activations (Shazeer, 2020)—are primarily introduced to stabilize training, and in some cases to increase expressivity. Understanding optimization from a principled perspective is therefore central to the design of scalable Transformer architectures.

Classical optimization theory (Nesterov, 1983; 1998; Nocedal & Wright, 1999; Boyd & Vandenberghe, 2004) provides a precise connection between optimization stability and second-order geometry. For a twice-differentiable objective $\ell(\boldsymbol{x})$, the local curvature is characterized by the Hessian $\boldsymbol{H_{xx}} = \nabla^2\ell(\boldsymbol{x})$. If $\ell$ is $\beta$-smooth, then $\|\nabla\ell(\boldsymbol{x}) - \nabla\ell(\boldsymbol{y})\|_2 \leq \beta\|\boldsymbol{x} - \boldsymbol{y}\|_2, \forall \boldsymbol{x}, \boldsymbol{y}$. Standard results imply that gradient descent is stable only when **the maximum tolerable learning rate** $\eta$ satisfies

$$\eta \leq \frac{2}{\beta} = \frac{2}{\sup_{\boldsymbol{x}} \|\boldsymbol{H_{xx}}\|_2},$$

establishing the Hessian spectral norm as the fundamental quantity governing admissible learning rates and convergence behavior.

In contrast, much of the recent literature on Transformer optimization focuses on architectural heuristics without explicitly analyzing their relationship with classical optimization theory. Techniques such as normalization placement (Vaswani et al., 2017; Wang et al., 2019; Henry et al., 2020; Qi et al., 2023a; 2025c;a), residual scaling (He et al., 2016; Bachlechner et al., 2021; Xie et al., 2025), or modified nonlinearities (Hendrycks, 2016; Shazeer, 2020) are typically justified empirically, while their impact on activation scale, Hessian geometry, and thereby optimal learning rates remains implicit. As a result, the theoretical relationship between network design and classical stability conditions is not well understood, despite its relevance to training very deep and large-scale models.

In this work, we bridge this gap by analyzing Transformer architectures through the central lens of Hessian-based optimization theory, while accounting for the role of activation scale. We introduce a simple normalization strategy, termed *SimpleNorm*, which stabilizes intermediate activation scales through normalization *immediately* following linear mappings. Building on this structural property, we analyze the Hessian of the loss with respect to network activations and

---

[*]Equal contribution [1]Tsinghua University, marco.ty.chen@gmail.com [2]Intellifusion Inc.. Correspondence to: Xianbiao Qi <qixianbiao@gmail.com>, Rong Xiao <rongxiao@gmail.com>.

*Proceedings of the 43rd International Conference on Machine Learning*, Seoul, South Korea. PMLR 306, 2026. Copyright 2026 by the author(s).

show that SimpleNorm significantly reduces the spectral norm of the Hessian, thereby permitting substantially larger stable learning rates. By grounding Transformer design in classical optimization principles (Nesterov, 1983; 1998; 2013), our framework provides a unified explanation for existing stabilization techniques and offers principled guidance for building scalable and stable models.

Our contributions can be summarized as follows:

- We revisit Transformer optimization through the lens of second-order geometry and establish a direct connection between architectural design, activation scale, the Hessian, and the maximum tolerable learning rate.

- We introduce SimpleGPT, a new GPT architecture based on SimpleNorm, and theoretically show that this design significantly reduces $\|\boldsymbol{H}_{xx}\|_2$, yielding a smaller Lipschitz gradient constant and enabling substantially larger stable learning rates.

- We demonstrate experimentally that these theoretical advantages are accompanied by consistent empirical gains across nanoGPT, Llama2, and Llama3 architectures, for model sizes ranging from 1B to 8B parameters. Specifically, when training 7B-scale models for 60K steps, our method achieves a training loss that is 0.08 lower than that of Llama2 with QKNorm, reducing the loss from 2.290 to 2.208. SimpleGPT also improves held-out language modeling performance and remains competitive on downstream tasks.

**Conflict of Interest Disclosure.** The authors declare no financial interests beyond the disclosed affiliations.

## 2. Related Work

**Normalization Methods.** Normalization has long been a central tool for stabilizing optimization and improving convergence in deep networks. Batch Normalization (BN) (Ioffe & Szegedy, 2015) normalizes activations using mini-batch statistics and has been widely successful in convolutional architectures, but its behavior can depend on batch size and distributed synchronization. Layer Normalization (LN) (Ba et al., 2016) and its variants remove batch dependence by computing statistics across features within each sample and have become the standard in Transformers. Related methods such as Instance Normalization (IN) (Ulyanov et al., 2016), Group Normalization (GN) (Wu & He, 2018), RMSNorm (Zhang & Sennrich, 2019), and nGPT (Loshchilov et al., 2025) further tailor normalization to specific architectural or efficiency constraints.

**Normalization Placement in Transformers.** Beyond the choice of normalization operator, its *placement* within Transformer blocks plays a critical role in optimization stability. The original Transformer architecture adopted post-normalization (PostNorm), in which normalization follows residual addition (Vaswani et al., 2017). Subsequent large-scale practice shifted toward pre-normalization (PreNorm), placing normalization before attention and MLP sublayers to improve trainability in deep networks (Wang et al., 2019).

Recent work further systematizes normalization placement and explores additional insertion points. Deeply Normalized Transformer (DNT) (Qi et al., 2025a) categorizes multiple strategies—including InputNorm, PreNorm, MidNorm, PostNorm, and QKNorm—and motivates them through a Jacobian- and gradient-stability analysis. DNT ultimately combines InputNorm, PreNorm, MidNorm, and QKNorm, while avoiding PostNorm due to its potential training instabilities. Among these placements, QK normalization (QKNorm) (Henry et al., 2020) specifically targets the attention mechanism, stabilizing the geometry of query–key interactions and mitigating softmax saturation.

By treating normalization as a design space over both operator and location, these works emphasize that stability and conditioning can be targeted at specific architectural subcomponents, rather than only at block outputs. As model depth increases, normalization also interacts with residual pathways and initialization. DeepNorm (Wang et al., 2022), for example, modifies residual scaling and initialization to bound parameter updates and control dynamical growth with depth, complementing normalization-placement strategies.

**Normalization-Free Transformers.** Motivated by the cost/complexity of normalization and the desire for simpler training dynamics, recent work questions whether explicit normalization is necessary in Transformers. *Transformers without Normalization* shows that replacing normalization layers with a simple point-wise nonlinearity, Dynamic Tanh (DyT) (Zhu et al., 2025), can match normalized baselines across tasks, suggesting that an appropriate bounded nonlinearity can provide much of the stability typically attributed to LN/RMSNorm. Building on this, *Stronger Normalization-Free Transformers* (Chen et al., 2025) studies the design of point-wise functions more broadly and reports improved normalization-free performance via a searched function family (e.g., Derf), outperforming LN/RMSNorm/DyT across multiple domains. Despite being framed as normalization-free, these approaches fundamentally operate by controlling the norm of activations through bounded transformations, and can therefore be viewed as a form of implicit normalization.

**Positioning of Our Work.** While our method can be viewed as a study of normalization placement in Transformers, its key distinction lies in explicitly linking architectural design to second-order optimization geometry. Rather than motivating normalization heuristically or empirically, we analyze how local normalization immediately following linear map-

pings stabilizes activation scale and, in turn, constrains the spectral norm of the Hessian and leads to a smoother optimization landscape. This perspective yields a principled characterization of the maximum tolerable learning rate and provides a unified theoretical explanation for optimization stability in large Transformer models.

## 3. Preliminaries

We consider the unconstrained convex optimization problem $\min_{\boldsymbol{x} \in \mathbb{R}^d} f(\boldsymbol{x})$, where $f : \mathbb{R}^d \to \mathbb{R}$ is *differentiable*.

### 3.1. Convex and Smoothed Optimization

**Lipschitz Gradient Smoothness.** If $f$ is twice differentiable, its second-order Taylor expansion at point $\boldsymbol{x}$ is

$$f(\boldsymbol{y}) \approx f(\boldsymbol{x}) + \langle \nabla f(\boldsymbol{x}), \boldsymbol{y} - \boldsymbol{x} \rangle \\ + \frac{1}{2}(\boldsymbol{y} - \boldsymbol{x})^\top \nabla^2 f(\boldsymbol{x})(\boldsymbol{y} - \boldsymbol{x}).$$

The second-order term captures the local curvature.

**Definition 3.1** ($\beta$-smoothness). The function $f$ is said to be $\beta$-smooth if

$$\|\nabla f(\boldsymbol{y}) - \nabla f(\boldsymbol{x})\|_2 \leq \beta \|\boldsymbol{y} - \boldsymbol{x}\|_2, \forall \boldsymbol{y}, \boldsymbol{x}.$$

For convex and differentiable functions, $\beta$-smoothness is equivalent to the following quadratic upper bound:

$$f(\boldsymbol{y}) \leq f(\boldsymbol{x}) + \langle \nabla f(\boldsymbol{x}), \boldsymbol{y} - \boldsymbol{x} \rangle + \frac{\beta}{2}\|\boldsymbol{y} - \boldsymbol{x}\|_2^2, \forall \boldsymbol{y}, \boldsymbol{x}.$$

This inequality plays a central role in step-size (learning rate) selection for optimization methods.

**Gradient Descent and Learning Rate.** Consider the standard gradient descent iteration

$$\boldsymbol{x}_{k+1} = \boldsymbol{x}_k - \eta \nabla f(\boldsymbol{x}_k),$$

where $\eta > 0$ is the learning rate. We wish to understand how the choice of $\eta$ depends on the smoothness constant $\beta$, and how this choice affects convergence.

**Descent Condition.** Evaluating the quadratic upper bound with $\boldsymbol{y} = \boldsymbol{x}_{k+1} = \boldsymbol{x}_k - \eta \nabla f(\boldsymbol{x}_k)$ and $\boldsymbol{x} = \boldsymbol{x}_k$, we obtain

$$f(\boldsymbol{x}_{k+1}) \leq f(\boldsymbol{x}_k) - \eta \|\nabla f(\boldsymbol{x}_k)\|_2^2 + \frac{\beta \eta^2}{2}\|\nabla f(\boldsymbol{x}_k)\|_2^2.$$

Rearranging terms gives

$$f(\boldsymbol{x}_{k+1}) \leq f(\boldsymbol{x}_k) - \left(\eta - \frac{\beta \eta^2}{2}\right)\|\nabla f(\boldsymbol{x}_k)\|_2^2.$$

A sufficient condition for monotone decrease of the objective is therefore

$$0 < \eta \leq \frac{2}{\beta}. \tag{1}$$

This bound characterizes the *stability region* of gradient descent for convex $\beta$-smooth functions. $\frac{2}{\beta}$ is usually known as the maximum tolerable learning rate.

For convex $\beta$-smooth problems, among all fixed learning rates that ensure descent, the canonical choice is typically $\eta = \frac{1}{\beta}$. This choice balances progress and stability and leads to the sharpest worst-case guarantees.

### 3.2. Gradient and Hessian of a Linear Projection

Given linear projection $\boldsymbol{y} = \boldsymbol{W}\boldsymbol{x}$ and loss function $\ell$, the gradient and Hessian of $\ell$ with respect to $\boldsymbol{y}$ are,

$$\boldsymbol{g_y} := \frac{\partial \ell}{\partial \boldsymbol{y}}^\top, \qquad \boldsymbol{H_{yy}} := \frac{\partial^2 \ell}{\partial \boldsymbol{y} \partial \boldsymbol{y}^\top}.$$

The Jacobian of $\boldsymbol{y}$ with respect to $\boldsymbol{x}$ is $\boldsymbol{J_x^y} = \frac{\partial \boldsymbol{y}}{\partial \boldsymbol{x}} = \boldsymbol{W}$. Hence, according to the chain rule, we have

$$\boldsymbol{g_x} = \frac{\partial \ell}{\partial \boldsymbol{x}}^\top = \boldsymbol{J_x^y}^\top \boldsymbol{g_y} = \boldsymbol{W}^\top \boldsymbol{g_y},$$

and the Hessian matrix with respect to $\boldsymbol{x}$ is

$$\boldsymbol{H_{xx}} = \frac{\partial^2 \ell}{\partial \boldsymbol{x} \partial \boldsymbol{x}^\top} = \boldsymbol{J_x^y}^\top \boldsymbol{H_{yy}} \boldsymbol{J_x^y} = \boldsymbol{W}^\top \boldsymbol{H_{yy}} \boldsymbol{W}. \tag{2}$$

## 4. Methodology

### 4.1. SimpleNorm: A Unified Normalization Strategy

**Definition of SimpleNorm.** We define *SimpleNorm* as placing a normalization operator *immediately* after a linear mapping. Given an input vector $\boldsymbol{x} \in \mathbb{R}^m$ and a linear transformation $\boldsymbol{W} \in \mathbb{R}^{d \times m}$, we abstract SimpleNorm as a primitive operator

$$\boldsymbol{\Psi}(\boldsymbol{x}) = \mathrm{Norm}(\boldsymbol{W}\boldsymbol{x}), \tag{3}$$

where $\mathrm{Norm}(\cdot)$ is a normalization operator such as LayerNorm or RMSNorm. SimpleNorm is motivated by a simple yet effective *placement* strategy, rather than algebraic complexity. In contrast to existing normalization techniques that typically operate at the level of residual blocks, hidden states, or parameter reparameterization, SimpleNorm enforces normalization *locally and immediately* after linear mapping, treating "linear mapping immediately followed by a normalization" as a single, unified operator.

**Definition of SimpleGPT.** As illustrated in Figure 1, *SimpleGPT* uses SimpleNorm as a fundamental building block. SimpleNorm is systematically inserted wherever a linear

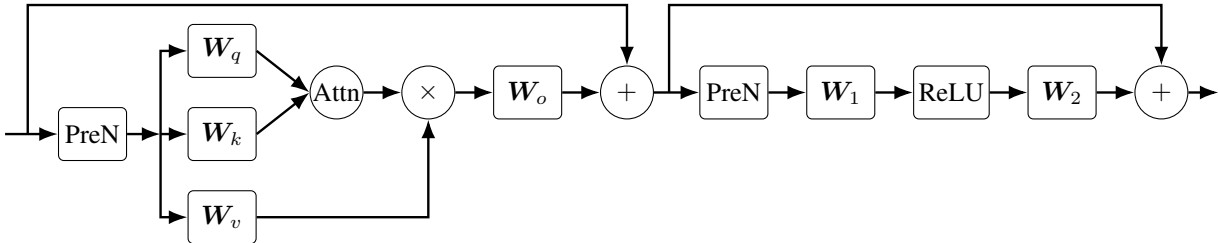

*(a)* GPT. GPT adopts a pre-normalization architecture. Linear projections $\boldsymbol{W}_q, \boldsymbol{W}_k, \boldsymbol{W}_v, \boldsymbol{W}_o, \boldsymbol{W}_1$ and $\boldsymbol{W}_2$ are applied.

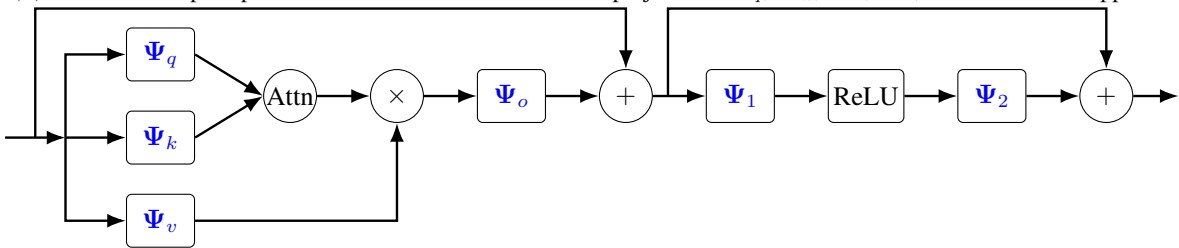

*(b)* SimpleGPT. SimpleGPT replaces all linear layers with **SimpleNorm** operator, denoted by $\boldsymbol{\Psi}$. In SimpleGPT, we do not use prenorm.

*Figure 1.* SimpleGPT vs. GPT. This figure compares the standard GPT block with the proposed SimpleGPT block, highlighting the structural simplifications introduced by SimpleNorm.

layer appears, including MLP projections, attention projections (Q, K, V), output projections, and gating or memory-related modules. Take Figure 1 as an example, normalization is inserted after the $\boldsymbol{W}_q, \boldsymbol{W}_k, \boldsymbol{W}_v, \boldsymbol{W}_o, \boldsymbol{W}_1$, and $\boldsymbol{W}_2$ projections. In architectures that employ SwiGLU (Shazeer, 2020) instead of MLP, SimpleGPT inserts normalization after $\boldsymbol{W}_q, \boldsymbol{W}_k, \boldsymbol{W}_v, \boldsymbol{W}_o, \boldsymbol{W}_1, \boldsymbol{W}_2$, and $\boldsymbol{W}_3$.

**Instantiating SimpleNorm with RMSNorm** In this work, we instantiate $\mathrm{Norm}(\cdot)$ with RMSNorm (Zhang & Sennrich, 2019). Hence, SimpleNorm is now defined as:

$$\boldsymbol{\Psi}(\boldsymbol{x}; \boldsymbol{W}, \boldsymbol{\gamma}) = \boldsymbol{\gamma} \odot \sqrt{d} \frac{\boldsymbol{W}\boldsymbol{x}}{\|\boldsymbol{W}\boldsymbol{x}\|_2}, \qquad (4)$$

where $\boldsymbol{x} \in \mathbb{R}^m$, $\boldsymbol{W} \in \mathbb{R}^{d \times m}$, and $\boldsymbol{W}, \boldsymbol{\gamma}$ are learnable parameters, and $\odot$ denotes element-wise multiply.

For later analysis, we define the intermediate variables

$$\boldsymbol{z} = \boldsymbol{W}\boldsymbol{x}, \qquad s = \|\boldsymbol{z}\|_2, \qquad \boldsymbol{u} = \frac{\boldsymbol{z}}{s},$$

$$\boldsymbol{P} = \boldsymbol{I} - \boldsymbol{u}\boldsymbol{u}^\top, \quad \boldsymbol{D} = \mathrm{Diag}(\boldsymbol{\gamma}),$$

so that $\boldsymbol{\Psi}(\boldsymbol{x}; \boldsymbol{W}, \boldsymbol{\gamma}) = \sqrt{d}\,\boldsymbol{D}\boldsymbol{u}$.

**Core Properties of SimpleNorm** The following sections prove two important mechanisms of SimpleNorm: SimpleNorm directly stabilizes the scale of activations to be on the order of $\sqrt{d}$, and SimpleNorm constrains the spectral norm of the Hessian of the loss w.r.t. the activations, smoothing the loss landscape and enabling larger learning rates. Moreover, we present a hypothesis for why, in ad-

dition to the predicted optimization stability, SimpleNorm also exhibits strong empirical performance.

### 4.2. Mechanism I: Stable Activation Scale

By construction, SimpleNorm stabilizes the scale of intermediate activations by normalizing *immediately* after each linear mapping. Recall that

$$\boldsymbol{\Psi}(\boldsymbol{x}; \boldsymbol{W}, \boldsymbol{\gamma}) = \sqrt{d}\,\boldsymbol{D}\boldsymbol{u} \qquad \text{with } \|\boldsymbol{u}\|_2 = 1.$$

Then

$$\|\boldsymbol{\Psi}(\boldsymbol{x}; \boldsymbol{W}, \boldsymbol{\gamma})\|_2 = \sqrt{d}\,\|\boldsymbol{D}\boldsymbol{u}\|_2.$$

In particular, letting $\gamma_{\min} = \min_i |\gamma_i|$ and $\gamma_{\max} = \max_i |\gamma_i|$, we have the bound

$$\gamma_{\min}\sqrt{d} \leq \|\boldsymbol{\Psi}(\boldsymbol{x})\|_2 \leq \gamma_{\max}\sqrt{d}.$$

Thus, up to the learned per-dimension scaling $\boldsymbol{\gamma}$, each projection is rescaled to have norm on the order of $\sqrt{d}$. Consequently, SimpleNorm prevents intermediate representation norms from drifting with depth or weight growth, eliminating a common source of activation explosion. Mechanism II shows how, in addition to activations, SimpleNorm also stabilizes curvature via the gradient Lipschitz constant.

### 4.3. Mechanism II: Smoother Loss Landscape

**Smoothness and the Hessian.** The smoothness of the objective directly constrains optimization stability: larger curvature implies smaller safe learning rates. Given a twice-differentiable $\beta$-smooth objective $\ell(x)$, we quantify local curvature by the activation Hessian $\boldsymbol{H}_{\boldsymbol{x}\boldsymbol{x}} = \nabla^2_{\boldsymbol{x}\boldsymbol{x}}\ell$. Specifically, the supremum of the spectral norm of the Hessian

upper bounds local curvature and governs gradient stability:

$$\beta = \sup_{\boldsymbol{x}} \|\boldsymbol{H}_{\boldsymbol{xx}}(\boldsymbol{x})\|_2.$$

To show SimpleNorm yields a smoother landscape, we prove two results: (i) the SimpleNorm Hessian decomposes as $\boldsymbol{H}_{\boldsymbol{xx}} = \boldsymbol{L} + \boldsymbol{C}$ and in high dimension $\|\boldsymbol{C}\|_2 \ll \|\boldsymbol{L}\|_2$; (ii) compared to a linear projection whose curvature scales as $\|\boldsymbol{W}\|_2^2$, the SimpleNorm curvature is scale-invariant with respect to $\|\boldsymbol{W}\|_2$. Combined, these imply $\|\boldsymbol{H}_{\boldsymbol{xx}}^{\mathrm{sn}}\|_2 \ll \|\boldsymbol{H}_{\boldsymbol{xx}}^{\mathrm{lin}}\|_2$ since $\|\boldsymbol{W}\|_2$ generally grows during training.

**SimpleNorm Derivatives.** First, we compute the first-order gradient and second-order Hessian of $\ell$ with respect to $\boldsymbol{x}$.

Given $\boldsymbol{y} = \sqrt{d}\,\boldsymbol{D}\boldsymbol{u}$, let

$$\ell = \ell(\boldsymbol{y}), \qquad \boldsymbol{g}_{\boldsymbol{y}} := \nabla_{\boldsymbol{y}}\ell, \qquad \boldsymbol{H}_{\boldsymbol{yy}} := \nabla^2_{\boldsymbol{yy}}\ell.$$

*First-order derivative.* The Jacobian of the normalization satisfies $\frac{\partial \boldsymbol{u}}{\partial \boldsymbol{z}} = \frac{1}{s}\boldsymbol{P}$, which yields the Jacobian of $\boldsymbol{y}$ with respect to $\boldsymbol{x}$:

$$\boldsymbol{J}_{\boldsymbol{x}}^{\boldsymbol{y}} := \frac{\partial \boldsymbol{y}}{\partial \boldsymbol{x}} = \frac{\sqrt{d}}{s}\,\boldsymbol{D}\,\boldsymbol{P}\,\boldsymbol{W}.$$

Applying the chain rule, the gradient is

$$\nabla_{\boldsymbol{x}}\ell = \boldsymbol{J}_{\boldsymbol{x}}^{\boldsymbol{y}\top}\boldsymbol{g}_{\boldsymbol{y}} = \frac{\sqrt{d}}{s}\,\boldsymbol{W}^\top \boldsymbol{P}\,\boldsymbol{D}\,\boldsymbol{g}_{\boldsymbol{y}}. \tag{5}$$

*Second-order derivative.* Differentiating the gradient leads to the standard decomposition

$$\boldsymbol{H}_{\boldsymbol{xx}} = \nabla^2_{\boldsymbol{x}}\ell = \underbrace{\boldsymbol{J}_{\boldsymbol{x}}^{\boldsymbol{y}\top}\boldsymbol{H}_{\boldsymbol{yy}}\boldsymbol{J}_{\boldsymbol{x}}^{\boldsymbol{y}}}_{\text{Gauss–Newton term}} + \underbrace{\boldsymbol{C}}_{\text{curvature term}}, \tag{6}$$

where the first term is the Gauss–Newton component

$$\boldsymbol{J}_{\boldsymbol{x}}^{\boldsymbol{y}\top}\boldsymbol{H}_{\boldsymbol{yy}}\boldsymbol{J}_{\boldsymbol{x}}^{\boldsymbol{y}} = \frac{d}{s^2}\,\boldsymbol{W}^\top \boldsymbol{P}\,\boldsymbol{D}\,\boldsymbol{H}_{\boldsymbol{yy}}\,\boldsymbol{D}\,\boldsymbol{P}\,\boldsymbol{W}, \tag{7}$$

and the second term is from the curvature of the normalization,

$$\boldsymbol{C} = -\frac{\sqrt{d}}{s^2}\,\boldsymbol{W}^\top \Big(\boldsymbol{P}\boldsymbol{D}\boldsymbol{g}_{\boldsymbol{y}}\boldsymbol{u}^\top + \boldsymbol{u}^\top \boldsymbol{D}\boldsymbol{g}_{\boldsymbol{y}}\boldsymbol{P} + \boldsymbol{u}\boldsymbol{g}_{\boldsymbol{y}}^\top \boldsymbol{D}\boldsymbol{P}\Big)\boldsymbol{W}. \tag{8}$$

Please see Appendix A for a detailed derivation of $\nabla_{\boldsymbol{x}}\ell$ and $\nabla^2_{\boldsymbol{x}}\ell$ for SimpleNorm. For completeness, derivations of $\nabla_{\boldsymbol{\gamma}}\ell$, $\nabla^2_{\boldsymbol{\gamma}}\ell$, $\nabla_{\boldsymbol{W}}\ell$ and $\nabla^2_{\mathrm{vec}(\boldsymbol{W})}\ell$ with $\boldsymbol{y} = \boldsymbol{\gamma} \odot \sqrt{d}\,\frac{\boldsymbol{W}\boldsymbol{x}}{\|\boldsymbol{W}\boldsymbol{x}\|_2}$ and $\nabla_{\boldsymbol{W}}\ell$ and $\nabla^2_{\mathrm{vec}(\boldsymbol{W})}\ell$ with $\boldsymbol{y} = \boldsymbol{W}\boldsymbol{x}$ are also provided in Appendix C and Appendix B

**Gauss–Newton Term Dominates in High Dimension.** Next, we show that under standard high-dimensional and non-pathological conditions, the Gauss–Newton term dominates the curvature induced by normalization.

**Theorem 4.1** (Gauss–Newton dominance for SimpleNorm). *Let $\boldsymbol{H}_{\boldsymbol{xx}} = \nabla^2_{\boldsymbol{x}}\ell$ denote the activation Hessian induced by SimpleNorm for a twice-differentiable objective $\ell(\boldsymbol{y})$. Then, the Hessian decomposes as*

$$\boldsymbol{H}_{\boldsymbol{xx}} = \boldsymbol{L} + \boldsymbol{C}, \qquad \boldsymbol{L} = (\boldsymbol{J}_{\boldsymbol{x}}^{\boldsymbol{y}})^\top \boldsymbol{H}_{\boldsymbol{yy}}\boldsymbol{J}_{\boldsymbol{x}}^{\boldsymbol{y}},$$

*where $\boldsymbol{C}$ is the curvature term induced by normalization.*

*Assume $\|\boldsymbol{x}\|_2 = \sqrt{d}$, $\boldsymbol{D} = \boldsymbol{I}$, $\boldsymbol{W} \in \mathbb{R}^{d \times d}$ has high effective rank $\|\boldsymbol{W}\|_F^2/\|\boldsymbol{W}\|_2^2 \geq c\,d$, and the input and loss derivatives are not pathologically aligned with $\boldsymbol{W}$. Define*

$$\kappa := \left\| \frac{\sqrt{d}\,\boldsymbol{W}}{\|\boldsymbol{W}\boldsymbol{x}\|_2} \right\|_2.$$

*Then $\kappa = \Theta(1)$ with high probability, and there exists a constant $\tau = \Theta(1)$ such that*

$$\|\boldsymbol{L}\|_2 = \tau\,\kappa^2\,\|\boldsymbol{H}_{\boldsymbol{yy}}\|_2, \qquad \|\boldsymbol{C}\|_2 \leq \frac{3\kappa^2}{\sqrt{d}}\,\|\boldsymbol{g}_{\boldsymbol{y}}\|_2.$$

*In particular, if $\|\boldsymbol{g}_{\boldsymbol{y}}\|_2 = O(\|\boldsymbol{H}_{\boldsymbol{yy}}\|_2)$, then*

$$\|\boldsymbol{C}\|_2 \ll \|\boldsymbol{L}\|_2$$

*so the Gauss–Newton term dominates the Hessian w.h.p.*

A complete proof is given in Appendix D. Although Theorem 4.1 is stated under the simplifying assumption $\boldsymbol{D} = \boldsymbol{I}$, we empirically verify that the learned RMSNorm gain parameters remain in a moderate range during training; see Appendix F for more details.

**SimpleNorm Hessian is Weight Scale-Invariant.** Finally, we compare the SimpleNorm Hessian's magnitude to that of a plain linear projection. We show that linear curvature grows quadratically with the weight matrix spectral norm $\|\boldsymbol{W}\|_2$, whereas SimpleNorm removes this dependence.

**Theorem 4.2** (Linear curvature scales with $\|\boldsymbol{W}\|_2^2$ while SimpleNorm does not). *Let $\ell = \ell(\boldsymbol{y})$ be twice differentiable, with $\boldsymbol{H}_{\boldsymbol{yy}} = \nabla^2_{\boldsymbol{yy}}\ell$ and $\boldsymbol{g}_{\boldsymbol{y}} = \nabla_{\boldsymbol{y}}\ell$.*

*Consider the linear mapping with its Hessian*

$$\boldsymbol{y}_1 = \boldsymbol{W}_1\boldsymbol{x}, \qquad \boldsymbol{H}_{xx}^{\mathrm{lin}} = \boldsymbol{W}_1^\top \boldsymbol{H}_{yy}\boldsymbol{W}_1,$$

*and the SimpleNorm mapping with its Hessian*

$$\boldsymbol{y}_2 = \boldsymbol{D}\frac{\sqrt{d}\,\boldsymbol{W}_2\boldsymbol{x}}{\|\boldsymbol{W}_2\boldsymbol{x}\|_2}, \qquad \boldsymbol{H}_{xx}^{\mathrm{sn}} = \boldsymbol{L} + \boldsymbol{C}.$$

*Assume $\boldsymbol{H}_{y_1 y_1} = \boldsymbol{H}_{y_2 y_2} := \boldsymbol{H}_{yy}$, $\boldsymbol{W}_1 = \boldsymbol{W}_2 := \boldsymbol{W}$, and that the conditions of Theorem 4.1 hold, such that $\|\boldsymbol{L}\|_2 \gg \|\boldsymbol{C}\|_2$. Then, with high-probability,*

$$\|\boldsymbol{H}_{\boldsymbol{xx}}^{\mathrm{sn}}\|_2 = \Theta\big(\kappa^2\,\|\boldsymbol{H}_{\boldsymbol{yy}}\|_2\big), \qquad \kappa^2 = \frac{d}{\|\widetilde{\boldsymbol{W}}\boldsymbol{x}\|_2^2} = \Theta(1),$$

*where $\widetilde{\boldsymbol{W}} = \boldsymbol{W}/\|\boldsymbol{W}\|_2$.*

*Moreover, if the range of $\widetilde{\boldsymbol{W}}$ is not adversarially aligned with the leading eigenspace of $\boldsymbol{H_{yy}}$, then there exists a constant $c_{\mathrm{lin}} = \Theta(1)$ such that*

$$\|\boldsymbol{H}_{\boldsymbol{xx}}^{\mathrm{lin}}\|_2 = \|\boldsymbol{W}^\top \boldsymbol{H_{yy}} \boldsymbol{W}\|_2 \ \geq \ c_{\mathrm{lin}} \|\boldsymbol{W}\|_2^2 \|\boldsymbol{H_{yy}}\|_2.$$

*Consequently, as $\|\boldsymbol{W}\|_2$ grows during training,*

$$\|\boldsymbol{H}_{\boldsymbol{xx}}^{\mathrm{lin}}\|_2 \ \gg \ \|\boldsymbol{H}_{\boldsymbol{xx}}^{\mathrm{sn}}\|_2 \qquad \textit{(with high probability)}.$$

Intuitively, SimpleNorm removes the dependence of curvature on weight scale by normalizing activations, whereas a linear projection amplifies curvature as $\|\boldsymbol{W}\|_2$ grows. We provide a proof for Theorem 4.2 in Appendix E.

**SimpleNorm Enables Larger Learning Rates.** For a twice-differentiable $\beta$-smooth objective, the maximum stable learning rate of gradient descent is inversely proportional to $\beta$, the Lipschitz constant of the gradient, which is equivalent to the supremum of the spectral norm of the Hessian: $\eta \leq \frac{2}{\beta}$ where $\beta = \sup_{\boldsymbol{x}} \|\boldsymbol{H}_{\boldsymbol{xx}}(\boldsymbol{x})\|_2$

Theorem 4.1 and Theorem 4.2 establish that, under standard high-dimensional and non-pathological conditions, the SimpleNorm Hessian is invariant to the spectral norm of the weight matrix whereas the Hessian of a linear projection scales quadratically with the weight norm.

Consequently, since the weight spectral norm generally grows throughout training, the SimpleNorm-based SimpleGPT architecture has a smoother loss landscape that can tolerate significantly larger learning rates compared to methods based on direct linear projections.

### 4.4. Interpretation: Beyond Optimization Stability

We have established two core properties of SimpleNorm: (i) it stabilizes activation scale at $\Theta(\sqrt{d})$, and (ii) it smooths the loss landscape by constraining the spectral norm of the activation Hessian, enabling larger and more stable learning rates. Although these properties explain the improved optimization stability of SimpleNorm, they do not fully account for the strong empirical performance observed in Section 5.

We hypothesize that SimpleNorm provides additional benefits at a more *global* representational level. By normalizing immediately after each linear projection, SimpleNorm ensures that every layer induces a genuinely nonlinear transformation, even in regimes where the surrounding network would otherwise behave nearly linearly. This effectively increases the depth of nonlinear interactions and enhances expressive capacity without increasing parameter count.

Under this view, SimpleNorm improves performance through a dual effect: locally, by improving optimization

geometry via reduced curvature variability; and globally, by increasing expressiveness through pervasive normalization-induced nonlinearity. We believe this combination explains why SimpleNorm yields consistent empirical gains beyond what would be expected from learning-rate stability alone.

### 4.5. Use `torch.compile` to Speedup Training

Normalization layers are memory-bound and frequently executed, making them a potential bottleneck. By fusing reduction and pointwise operations and leveraging `torch.compile`, SimpleNorm's increased normalization overhead is largely amortized, resulting in around a 3% training-time increase compared to GPT with QKNorm.

## 5. Experiments

**Experimental Settings.** We evaluate SimpleNorm on three Transformer backbones: nanoGPT, Llama2, and Llama3. SimpleNorm is applied to all Transformer blocks, excluding the embedding and output layers. All models are trained using the AdamW optimizer (Kingma & Ba, 2014; Loshchilov & Hutter, 2019) with cosine learning-rate scheduling with bfloat16 precision. Learning rates are tuned for each method. Since SimpleNorm permits significantly larger stable learning rates, we adjust weight decay accordingly. Additional architectural, hyperparameter, and training details are provided in Appendix G and Appendix H.

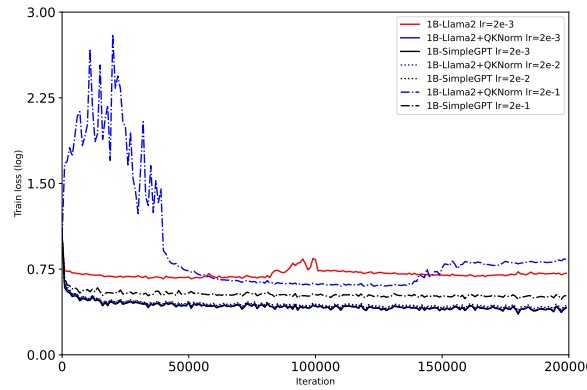

*Figure 2.* The largest admissible learning rate for Llama2-1B, Llama2-1B with QKNorm, and SimpleGPT-1B.

### 5.1. Largest Tolerable Learning Rate

We evaluate the largest tolerable learning rate by comparing optimization stability across different normalization schemes while keeping all other training settings fixed. As shown in Figure 2, PreNorm already exhibits convergence issues at a learning rate of $2 \times 10^{-3}$. In contrast, PreNorm+QKNorm remains stable at $2\times10^{-3}$ and $2\times10^{-2}$, but becomes unstable when the learning rate is increased to $2 \times 10^{-1}$. SimpleNorm shows stable convergence at both

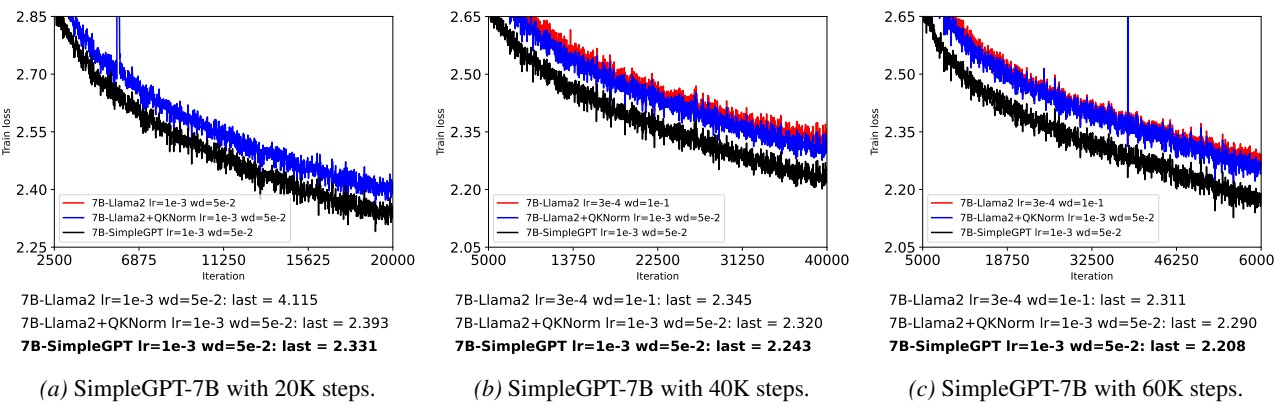

*(a)* SimpleGPT-7B with 20K steps.   *(b)* SimpleGPT-7B with 40K steps.   *(c)* SimpleGPT-7B with 60K steps.

*Figure 3.* The training loss curves of Llama2-7B, Llama2-7B with QKNorm and SimpleGPT-7B under 20K, 40K and 60K training steps.

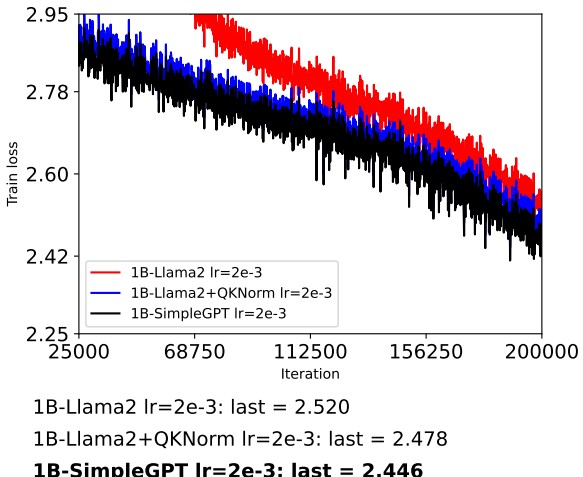

*Figure 4.* The training loss curves of Llama2-1B, Llama2-1B with QKNorm and SimpleGPT-1B under 200K training steps.

$2 \times 10^{-3}$ and $2 \times 10^{-2}$, and is notably more stable than PreNorm+QKNorm at $2 \times 10^{-1}$. Overall, these results suggest that SimpleNorm consistently tolerates larger learning rates, indicating improved optimization robustness.

### 5.2. SimpleGPT-1B Based on Llama2

In this subsection, we evaluate SimpleGPT-1B and compare it against the standard Llama2-1B as well as Llama2-1B with QKNorm. We train the model for 200K steps (following (Zhang et al., 2024)), with a global batch size of 256 and a sequence length of 512, resulting in approximately 26B training tokens. We train all models on the C4 dataset following the same training recipe as their corresponding baselines. The results are presented in Figure 4. The loss curve is smoothed by 80% in Tensorboard. For all experiments, we report the training loss of the last step.

In Figure 4, SimpleGPT-1B achieves notable improvement over Llama2-1B with QKNorm. Specifically, the training

loss is reduced from 2.478 to 2.446, corresponding to an absolute improvement of 0.032. Hence, SimpleGPT provides measurable gains, even at the small 1B scale.

### 5.3. SimpleGPT-7B Based on Llama2

We compare SimpleGPT-7B against the standard Llama2-7B and Llama2-7B with QKNorm in Figure 3. We train the models for 20K, 40K, and 60K steps, corresponding to approximately 8B, 16B, and 24B tokens, respectively. All models are trained on the C4 dataset following the same training recipe as their corresponding baselines. SimpleGPT-7B uses a 0.001 learning rate, which is $3\times$ larger than that used in the Llama2-7B (Touvron et al., 2023b) model.

We make the following observations. First, the performance gain of SimpleGPT over Llama2+QKNorm is consistently significant throughout training: the improvement reaches 0.062 at 20K steps, increases to 0.077 at 40K steps, and remains at a comparable level (0.082) at 60K steps. Second, SimpleGPT maintains more stable training dynamics compared to Llama2 with QKNorm. Third, as training progresses and more tokens are observed, the relative improvement does not diminish, indicating that the advantage of SimpleGPT is stable rather than a transient early-training effect. Finally, we observe a clear scaling trend with respect to model size. While the 1B model trained on 26B tokens achieves a modest improvement of approximately 0.03, the 7B model trained on 24B tokens exhibits a substantially larger gain of 0.08.

### 5.4. SimpleGPT-8B Based on Llama3

At the 8B scale, our experiments are based on the Llama3-8B architecture. We train both SimpleGPT-8B and Llama3-8B on the C4 dataset with a global batch size of 192 and a sequence length of 2048. We conduct training for 20K steps, corresponding to approximately 8B training tokens. We do not train for more steps due to compute constraints.

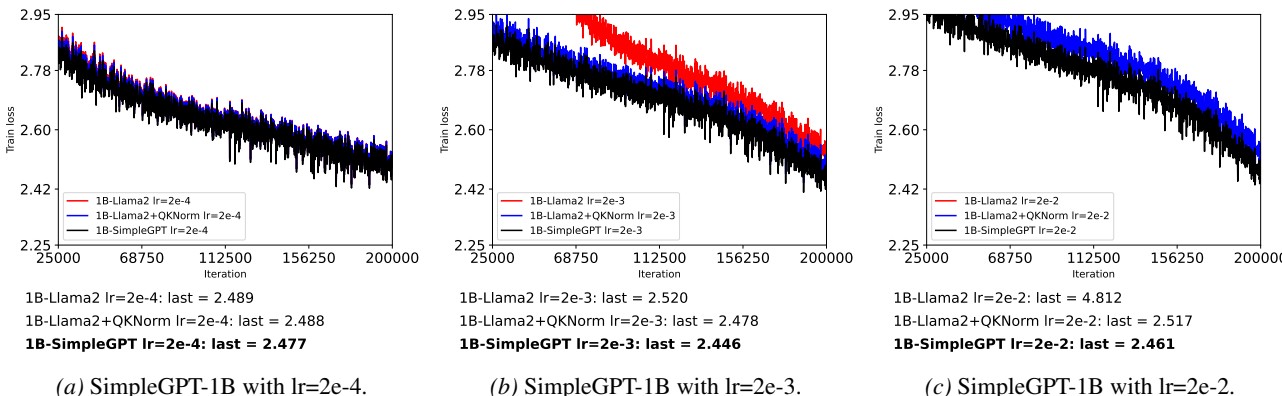

*(a)* SimpleGPT-1B with lr=2e-4.

*(b)* SimpleGPT-1B with lr=2e-3.

*(c)* SimpleGPT-1B with lr=2e-2.

*Figure 5.* Overall comparison across Llama2-1B, Llama2-1B with QKNorm and SimpleGPT-1B under three different learning rates. Adam-mini uses a $2 \times 10^{-4}$ learning rate. In SimpleGPT, we enable a $10\times$ learning rate and obtain better performance.

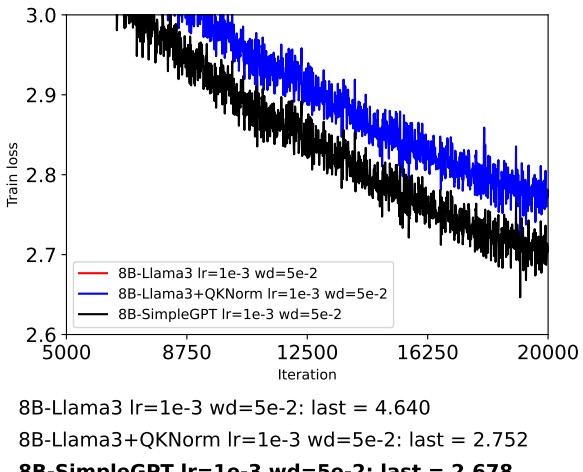

*Figure 6.* The training loss curves of Llama3-8B, Llama3-8B with QKNorm and SimpleGPT-8B.

SimpleGPT-8B employs a $3\times$ larger learning rate than Llama3-8B, and as shown in Figure 6, achieves a substantially lower training loss. Moreover, the magnitude of the performance gain is consistent with that observed for the 7B model, suggesting that our method exhibits favorable scaling behavior with increasing model size.

### 5.5. SimpleGPT-1.4B Based on nanoGPT

Finally, we evaluate SimpleGPT-1.4B on the nanoGPT code base. All models are trained for 100K steps, corresponding to approximately 50B tokens. SimpleGPT-1.4B is trained using a learning rate that is $3\times$ larger than the baseline.

We report validation losses in Figure 7. Note that, since validation loss is recorded once every 1,000 steps, the curves in Figure 7 appear different compared to earlier figures.

We observe that GPT-2 with QKNorm achieves nearly identical performance to the original GPT-2, indicating that

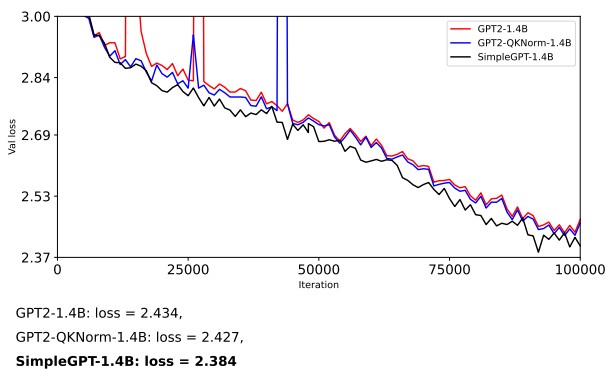

*Figure 7.* The validation loss curves of GPT2 1.4B, GPT2 1.4B with QKNorm and SimpleGPT 1.4B under 100K training steps.

QKNorm alone provides limited benefits in this setting. Consistent with the results on Llama2-1B, SimpleGPT-1.4B Based on nanoGPT yields an improvement of approximately 0.043. These findings suggest that the gains introduced by SimpleNorm are stable across architectures.

### 5.6. Held-Out and Downstream Evaluation

While the preceding experiments primarily evaluate optimization behavior through training and validation losses, we further examine whether the gains of SimpleGPT translate to held-out and downstream performance.

We compare SimpleGPT-7B with Llama2-7B and Llama2-7B with QKNorm. For held-out language modeling evaluation, we report bits-per-byte (bpb) on C4 validation and Wiki, where lower is better. For downstream evaluation, we report accuracy on WinoGrande and normalized accuracy on PIQA and HellaSwag, where higher is better.

As shown in Table 1, SimpleGPT-7B achieves lower bpb on both C4 validation and Wiki, indicating improved held-out language modeling performance. On downstream tasks, SimpleGPT-7B performs competitively with Llama2-7B

*Table 1.* Held-out and downstream evaluation of SimpleGPT-7B, Llama2-7B with QKNorm, and Llama2-7B. Lower is better for bpb, and higher is better for accuracy-based metrics.

| Dataset / Metric | SimpleGPT-7B | Llama2-7B + QKNorm | Llama2-7B |
|---|---|---|---|
| C4 val bpb ↓ | **0.7999** | 0.8327 | 0.8371 |
| Wiki bpb ↓ | **0.7896** | 0.8188 | 0.8227 |
| WinoGrande Acc. ↑ | **53.99** | 52.88 | 49.65 |
| PIQA Acc.-norm ↑ | 65.61 | **65.77** | 64.03 |
| HellaSwag Acc.-norm ↑ | **45.92** | 45.33 | 43.40 |

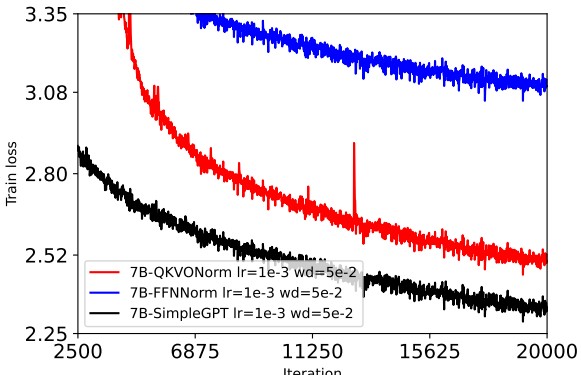

7B-QKVONorm lr=1e-3 wd=5e-2: last = 2.494

7B-FFNNorm lr=1e-3 wd=5e-2: last = 3.088

**7B-SimpleGPT lr=1e-3 wd=5e-2: last = 2.331**

*Figure 8.* Ablation on where SimpleNorm is applied in Llama2-7B. Comparison between full SimpleGPT, applying SimpleNorm only to the attention projections ($W_q, W_k, W_v, W_o$), and applying SimpleNorm only to the SwiGLU projections ($W_1, W_2, W_3$).

with QKNorm, achieving slightly higher scores on Wino-Grande and HellaSwag while remaining comparable on PIQA. These results suggest that the optimization improvements introduced by SimpleNorm are not merely reflected in training loss, but also lead to improved held-out performance and competitive downstream transfer.

### 5.7. Ablation Study and Additional Comparisons

**Learning-Rate Sensitivity.** As shown in Figure 5, we compare Llama2-1B, Llama2-1B with QKNorm, and SimpleGPT-1B under three learning-rate settings to assess whether the improvement of SimpleGPT is solely due to using a larger learning rate.

At the small learning rate $2 \times 10^{-4}$, all methods train stably, but SimpleGPT-1B achieves the lowest loss. As the learning rate increases to $2 \times 10^{-3}$ and $2 \times 10^{-2}$, the original Llama2-1B increasingly degrades, while QKNorm remains trainable and SimpleGPT consistently achieves the strongest loss. Overall, these results indicate that SimpleGPT provides gains even under matched learning rates and improves robustness to larger learning rates.

**Ablation on SimpleNorm Placement.** We further ablate where SimpleNorm should be applied within the Trans-

former block. Using Llama2-7B as the base architecture, we compare the full SimpleGPT design with two partial variants: one that applies SimpleNorm only to the attention projections ($W_q, W_k, W_v, W_o$), and one that applies SimpleNorm only to the SwiGLU projections ($W_1, W_2, W_3$).

As shown in Figure 8, the full SimpleGPT architecture is more stable and achieves significantly lower training loss than both partial variants. This suggests that the benefit of SimpleNorm does not come from a single isolated insertion point, but from systematically normalizing the outputs of all linear mappings across the block.

We also compare SimpleGPT with SandwichNorm (Ding et al., 2021) in Appendix I. SimpleGPT achieves lower training loss, further suggesting the importance of normalization placement immediately after linear mappings.

### 5.8. Practical Overhead of SimpleGPT

**Training Time.** We compare the training speed of our SimpleGPT-8B model with that of Llama3-8B with QKNorm. On average, Llama3-8B requires 1553 ms per training step while SimpleGPT-8B takes 1603 ms per step. This corresponds to a slowdown of around 3%, which can likely be further reduced through kernel-level fusion or more optimized normalization kernels.

**Memory Overhead.** We also measure the training memory footprint under the Llama2-7B setting on 8 A800 GPUs, with global batch configuration $8 \times 6 \times 4 \times 2048$ (GPUs, gradient accumulation steps, local batch size, and sequence length). The peak memory usage is 54.4 GB for Llama2-7B with PreNorm, 58.5 GB for Llama2-7B with PreNorm+QKNorm, and 60.5 GB for SimpleGPT-7B. Thus, SimpleGPT adds approximately 2.0 GB per GPU over QKNorm in this setting.

## 6. Conclusion

In this work, we revisit Transformer optimization from a second-order perspective and establish a direct connection between architectural design, the Hessian matrix, and optimization stability. By introducing SimpleNorm and analyzing its induced Hessian structure, we show that reducing the Hessian norm of the activation with respect to the loss enables substantially larger admissible learning rates. The resulting model, **SimpleGPT**, reliably supports learning rates up to 3×-10× larger than strong baselines while maintaining stable optimization. Across extensive experiments on nanoGPT, Llama2, and Llama3 style models, spanning parameter scales from 1B to 8B, our method consistently achieves substantially stronger performance than GPT with QKNorm. Importantly, these gains are obtained with modest additional training-time and memory overhead.

## Acknowledgments

This work was supported in part by the National Natural Science Foundation of China under Grant No. 62576048.

## Impact Statement

This paper presents work whose goal is to advance the field of Machine Learning. There are many potential societal consequences of our work, none which we feel must be specifically highlighted here.

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

# A. Derivatives of $\nabla_{\boldsymbol{x}} \ell$ and $\nabla_{\boldsymbol{x}}^2 \ell$ for SimpleNorm

We consider the mapping

$$\boldsymbol{y} = \boldsymbol{\gamma} \odot \sqrt{d} \, \frac{\boldsymbol{W} \boldsymbol{x}}{\|\boldsymbol{W} \boldsymbol{x}\|_2},$$

where $\boldsymbol{\gamma} \in \mathbb{R}^d$, and $\odot$ denotes elementwise multiplication and $\ell = \ell(\boldsymbol{y})$ is a scalar loss.

As before, let us define some intermediate variables,

$$\boldsymbol{z} = \boldsymbol{W} \boldsymbol{x}, \qquad s = \|\boldsymbol{z}\|_2, \qquad \boldsymbol{u} = \frac{\boldsymbol{z}}{s}, \qquad \boldsymbol{P} = \boldsymbol{I} - \boldsymbol{u} \boldsymbol{u}^\top, \qquad \boldsymbol{D} = \text{Diag}(\boldsymbol{\gamma}),$$

where $\boldsymbol{P}, \boldsymbol{D}$ are symmetric.

Let us define the normalized and scaled output as

$$\boldsymbol{y} = \sqrt{d} \, \boldsymbol{D} \, \boldsymbol{u},$$

and let

$$\boldsymbol{g}_{\boldsymbol{y}} := \nabla_{\boldsymbol{y}} \ell, \qquad \boldsymbol{H}_{\boldsymbol{y}\boldsymbol{y}} := \nabla_{\boldsymbol{y}\boldsymbol{y}}^2 \ell.$$

Let us define the Jacobian $\boldsymbol{A}$ of the normalization map $\boldsymbol{u}$ with respect to $\boldsymbol{z}$ as

$$\boldsymbol{A} := \frac{\partial \boldsymbol{u}}{\partial \boldsymbol{z}} = \frac{1}{s} \, \boldsymbol{P},$$

and consequently

$$\frac{\partial \boldsymbol{y}}{\partial \boldsymbol{z}} = \sqrt{d} \, \boldsymbol{D} \, \boldsymbol{A} = \frac{\sqrt{d}}{s} \, \boldsymbol{D} \, \boldsymbol{P}.$$

We will repeatedly use the chain rule through the path

$$\boldsymbol{W} \to \boldsymbol{z} \to \boldsymbol{u} \to \boldsymbol{y} \to \ell, \qquad \text{and similarly} \qquad \boldsymbol{\gamma} \to \boldsymbol{D} \to \boldsymbol{y} \to \ell.$$

**(1) First-order derivative.** Since $\boldsymbol{z} = \boldsymbol{W} \boldsymbol{x}$, we have

$$d\boldsymbol{z} = \boldsymbol{W} \, d\boldsymbol{x}.$$

From $\boldsymbol{u} = \boldsymbol{z}/s$ and $\boldsymbol{A} = \partial \boldsymbol{u}/\partial \boldsymbol{z}$, it follows that

$$d\boldsymbol{u} = \boldsymbol{A} \, d\boldsymbol{z} = \frac{1}{s} \, \boldsymbol{P} \, d\boldsymbol{z}.$$

Therefore,

$$d\boldsymbol{y} = \sqrt{d} \, \boldsymbol{D} \, d\boldsymbol{u} = \sqrt{d} \, \boldsymbol{D} \, \boldsymbol{A} \, \boldsymbol{W} \, d\boldsymbol{x},$$

so the Jacobian is

$$\boldsymbol{J}_{\boldsymbol{x}}^{\boldsymbol{y}} := \frac{\partial \boldsymbol{y}}{\partial \boldsymbol{x}} = \sqrt{d} \, \boldsymbol{D} \, \boldsymbol{A} \, \boldsymbol{W} = \frac{\sqrt{d}}{s} \, \boldsymbol{D} \, \boldsymbol{P} \, \boldsymbol{W}.$$

Applying the chain rule gives

$$\nabla_{\boldsymbol{x}} \ell = \boldsymbol{J}_{\boldsymbol{x}}^{\boldsymbol{y}\top} \boldsymbol{g}_{\boldsymbol{y}} = \sqrt{d} \, \boldsymbol{W}^\top \boldsymbol{A}^\top \boldsymbol{D}^\top \boldsymbol{g}_{\boldsymbol{y}}.$$

Using $\boldsymbol{A}^\top = \boldsymbol{A}$ and $\boldsymbol{D}^\top = \boldsymbol{D}$, we obtain

$$\nabla_{\boldsymbol{x}} \ell = \sqrt{d} \, \boldsymbol{W}^\top \boldsymbol{A} \, \boldsymbol{D} \, \boldsymbol{g}_{\boldsymbol{y}} = \frac{\sqrt{d}}{s} \, \boldsymbol{W}^\top \boldsymbol{P} \, \boldsymbol{D} \, \boldsymbol{g}_{\boldsymbol{y}}. \tag{9}$$

**(2) Second-order derivative.** Differentiating $\nabla_{\boldsymbol{x}}\ell = \sqrt{d}\,\boldsymbol{W}^\top\boldsymbol{A}\,\boldsymbol{D}\,\boldsymbol{g_y}$ yields

$$d(\nabla_{\boldsymbol{x}}\ell) = \sqrt{d}\,\boldsymbol{W}^\top\Big(\boldsymbol{A}\,\boldsymbol{D}\,d\boldsymbol{g_y} + d\boldsymbol{A}\,\boldsymbol{D}\,\boldsymbol{g_y}\Big).$$

This induces the standard decomposition

$$\nabla_{\boldsymbol{x}}^2\ell = \boldsymbol{J_x^y}^\top\boldsymbol{H_{yy}}\boldsymbol{J_x^y} + \boldsymbol{C},$$

where the two terms are computed below.

(A) Linear (Gauss–Newton) term.

Since $d\boldsymbol{g_y} = \boldsymbol{H_{yy}}\,d\boldsymbol{y}$, and $d\boldsymbol{y} = \boldsymbol{J_x^y}\,d\boldsymbol{x}$, we have

$$d\boldsymbol{g_y} = \boldsymbol{H_{yy}}\,\boldsymbol{J_x^y}\,d\boldsymbol{x}.$$

Substituting into $\sqrt{d}\,\boldsymbol{W}^\top\boldsymbol{A}\,\boldsymbol{D}\,d\boldsymbol{g_y}$ gives

$$\sqrt{d}\,\boldsymbol{W}^\top\boldsymbol{A}\,\boldsymbol{D}\,d\boldsymbol{g_y} = \sqrt{d}\,\boldsymbol{W}^\top\boldsymbol{A}\,\boldsymbol{D}\,\boldsymbol{H_{yy}}\,\boldsymbol{J_x^y}\,d\boldsymbol{x} = \boldsymbol{J_x^y}^\top\boldsymbol{H_{yy}}\boldsymbol{J_x^y}\,d\boldsymbol{x}.$$

Using $\boldsymbol{A} = \boldsymbol{P}/s$ and $\boldsymbol{J_x^y} = (\sqrt{d}/s)\boldsymbol{DPW}$, we obtain the explicit form

$$\boldsymbol{J_x^y}^\top\boldsymbol{H_{yy}}\boldsymbol{J_x^y} = \frac{d}{s^2}\,\boldsymbol{W}^\top\boldsymbol{P}\,\boldsymbol{D}\,\boldsymbol{H_{yy}}\,\boldsymbol{D}\,\boldsymbol{P}\,\boldsymbol{W}.$$

(B) Curvature (normalization) term.

Recall $\boldsymbol{A} = \frac{1}{s}\boldsymbol{P}$. Using $d\boldsymbol{u} = \boldsymbol{A}\,d\boldsymbol{z}$ and $ds = d\|\boldsymbol{z}\|_2 = \boldsymbol{u}^\top d\boldsymbol{z}$, one obtains

$$d\boldsymbol{A} = -\frac{1}{s^2}\Big((\boldsymbol{u}^\top d\boldsymbol{z})\boldsymbol{P} + \boldsymbol{P}\,d\boldsymbol{z}\,\boldsymbol{u}^\top + \boldsymbol{u}\,d\boldsymbol{z}^\top\boldsymbol{P}\Big).$$

Right-multiplying by $\boldsymbol{D}\,\boldsymbol{g_y}$ gives

$$d\boldsymbol{A}\,\boldsymbol{D}\,\boldsymbol{g_y} = -\frac{1}{s^2}\Big((\boldsymbol{u}^\top d\boldsymbol{z})\boldsymbol{P}\,\boldsymbol{D}\,\boldsymbol{g_y} + \boldsymbol{P}\,d\boldsymbol{z}\,\boldsymbol{u}^\top\boldsymbol{D}\,\boldsymbol{g_y} + \boldsymbol{u}\,d\boldsymbol{z}^\top\boldsymbol{P}\,\boldsymbol{D}\,\boldsymbol{g_y}\Big).$$

Equivalently, pulling the scalar contractions to the left yields the linear map in $d\boldsymbol{z}$:

$$d\boldsymbol{A}\,\boldsymbol{D}\,\boldsymbol{g_y} = -\frac{1}{s^2}\Big(\boldsymbol{P}\boldsymbol{D}\boldsymbol{g_y}\boldsymbol{u}^\top + \boldsymbol{u}^\top\boldsymbol{D}\boldsymbol{g_y}\boldsymbol{P} + \boldsymbol{u}\boldsymbol{g_y}^\top\boldsymbol{D}\boldsymbol{P}\Big)\,d\boldsymbol{z}.$$

Substituting $d\boldsymbol{z} = \boldsymbol{W}\,d\boldsymbol{x}$ and left-multiplying by $\sqrt{d}\,\boldsymbol{W}^\top$ gives the curvature term

$$\boldsymbol{C} = -\frac{\sqrt{d}}{s^2}\,\boldsymbol{W}^\top\Big(\boldsymbol{P}\boldsymbol{D}\boldsymbol{g_y}\boldsymbol{u}^\top + \boldsymbol{u}^\top\boldsymbol{D}\boldsymbol{g_y}\boldsymbol{P} + \boldsymbol{u}\boldsymbol{g_y}^\top\boldsymbol{D}\boldsymbol{P}\Big)\boldsymbol{W}.$$

(If desired, define $\boldsymbol{g_1} := \boldsymbol{D}\boldsymbol{g_y}$ to simplify the expression, but we keep the figure's symbols explicit.)

Combining the linear and curvature terms, the Hessian with respect to $\boldsymbol{x}$ is

$$\boldsymbol{H_{xx}} = \nabla_{\boldsymbol{x}}^2\ell = \frac{d}{s^2}\,\boldsymbol{W}^\top\boldsymbol{P}\,\boldsymbol{D}\,\boldsymbol{H_{yy}}\,\boldsymbol{D}\,\boldsymbol{P}\,\boldsymbol{W} - \frac{\sqrt{d}}{s^2}\,\boldsymbol{W}^\top\Big(\boldsymbol{P}\boldsymbol{D}\boldsymbol{g_y}\boldsymbol{u}^\top + \boldsymbol{u}^\top\boldsymbol{D}\boldsymbol{g_y}\boldsymbol{P} + \boldsymbol{u}\boldsymbol{g_y}^\top\boldsymbol{D}\boldsymbol{P}\Big)\boldsymbol{W}, \tag{10}$$

where $\boldsymbol{z} = \boldsymbol{W}\boldsymbol{x}, s = \|\boldsymbol{z}\|_2, \boldsymbol{u} = \frac{\boldsymbol{z}}{s}, \boldsymbol{P} = \boldsymbol{I} - \boldsymbol{u}\boldsymbol{u}^\top, \boldsymbol{D} = \mathrm{Diag}(\boldsymbol{\gamma})$.

# B. Derivatives of $\nabla_{\boldsymbol{W}}\ell$ and $\nabla^2_{\mathrm{vec}(\boldsymbol{W})}\ell$ for $\boldsymbol{y} = \boldsymbol{W}\boldsymbol{x}$

We consider the linear mapping

$$\boldsymbol{y} = \boldsymbol{W}\boldsymbol{x},$$

where $\boldsymbol{W} \in \mathbb{R}^{d \times m}$, $\boldsymbol{x} \in \mathbb{R}^m$, and $\boldsymbol{y} \in \mathbb{R}^d$. Let $\ell = \ell(\boldsymbol{y})$ be a scalar-valued loss function.

We define the first- and second-order derivatives of the loss $l$ with respect to $\boldsymbol{y}$ as

$$\boldsymbol{g_y} := \nabla_{\boldsymbol{y}}\ell(\boldsymbol{y}) \in \mathbb{R}^d, \qquad \boldsymbol{H_{yy}} := \nabla^2_{\boldsymbol{y}}\ell(\boldsymbol{y}) \in \mathbb{R}^{d \times d}.$$

**(1) First-order derivative with respect to $\boldsymbol{W}$.** Since $\boldsymbol{y}$ depends linearly on $\boldsymbol{W}$, a first-order variation satisfies

$$d\boldsymbol{y} = d\boldsymbol{W}\,\boldsymbol{x}.$$

Applying the chain rule,

$$d\ell = \boldsymbol{g_y}^\top d\boldsymbol{y} = \boldsymbol{g_y}^\top(d\boldsymbol{W}\,\boldsymbol{x}) = \langle \boldsymbol{g_y}\boldsymbol{x}^\top,\, d\boldsymbol{W}\rangle_F.$$

Matching coefficients under the Frobenius inner product yields the gradient

$$\nabla_{\boldsymbol{W}}\ell = \boldsymbol{g_y}\,\boldsymbol{x}^\top. \tag{11}$$

**(2) Second-order derivative with respect to $\boldsymbol{W}$.** To express second-order derivatives, it is convenient to vectorize $\boldsymbol{W}$. Using the standard identity for matrix–vector products,

$$\boldsymbol{y} = (\boldsymbol{x}^\top \otimes \boldsymbol{I}_d)\,\mathrm{vec}(\boldsymbol{W}),$$

which makes the dependence of $\boldsymbol{y}$ on $\mathrm{vec}(\boldsymbol{W})$ explicit.

**Jacobian with respect to $\mathrm{vec}(\boldsymbol{W})$.** From the above expression, the Jacobian of $\boldsymbol{y}$ with respect to $\mathrm{vec}(\boldsymbol{W})$ is

$$\boldsymbol{J}^{\boldsymbol{y}}_{\mathrm{vec}(\boldsymbol{W})} \triangleq \frac{\partial \boldsymbol{y}}{\partial\,\mathrm{vec}(\boldsymbol{W})} = \boldsymbol{x}^\top \otimes \boldsymbol{I}_d \in \mathbb{R}^{d \times (md)}.$$

**Hessian with respect to $\mathrm{vec}(\boldsymbol{W})$.** Because the mapping $\boldsymbol{W} \mapsto \boldsymbol{y}$ is linear, it contributes no second-order term. Hence, by the second-order chain rule, the Hessian of the loss with respect to $\mathrm{vec}(\boldsymbol{W})$ is

$$\nabla^2_{\mathrm{vec}(\boldsymbol{W})}\ell = \boldsymbol{J}^{\boldsymbol{y}}_{\mathrm{vec}(\boldsymbol{W})}{}^\top \boldsymbol{H_{yy}}\,\boldsymbol{J}^{\boldsymbol{y}}_{\mathrm{vec}(\boldsymbol{W})}.$$

Substituting the explicit Jacobian gives us

$$\nabla^2_{\mathrm{vec}(\boldsymbol{W})}\ell = (\boldsymbol{x} \otimes \boldsymbol{I}_d)\,\boldsymbol{H_{yy}}\,(\boldsymbol{x}^\top \otimes \boldsymbol{I}_d).$$

**Kronecker-product simplification.** Using standard identities of the Kronecker product, the Hessian simplifies to

$$\nabla^2_{\mathrm{vec}(\boldsymbol{W})}\ell = (\boldsymbol{x}\boldsymbol{x}^\top) \otimes \boldsymbol{H_{yy}}. \tag{12}$$

This result shows that the Hessian with respect to the weight matrix factorizes into a Kronecker product of an input-dependent term $\boldsymbol{x}\boldsymbol{x}^\top$ and an output-space curvature term $\boldsymbol{H_{yy}}$. Equivalently, when viewed as a block matrix with $m \times m$ blocks of size $d \times d$, the $(i,j)$-th block is

$$\left[\nabla^2_{\mathrm{vec}(\boldsymbol{W})}\ell\right]_{(i,j)\,\mathrm{block}} = x_i x_j\,\boldsymbol{H_{yy}},$$

where $x_i$ denotes the $i$-th entry of $\boldsymbol{x}$.

# C. Derivatives of $\nabla_\gamma \ell$, $\nabla_\gamma^2 \ell$, $\nabla_W \ell$ and $\nabla_{\text{vec}(W)}^2 \ell$ where $y = \gamma \odot \sqrt{d} \, \frac{Wx}{\|Wx\|_2}$.

We consider the mapping

$$y = \gamma \odot \sqrt{d} \, \frac{Wx}{\|Wx\|_2},$$

where $\odot$ denotes elementwise multiplication and $\ell = \ell(y)$ is a scalar loss. Define the intermediate variables

$$z = Wx, \qquad s = \|z\|_2, \qquad u = \frac{z}{s}, \qquad P = I - uu^\top, \qquad D = \text{Diag}(\gamma).$$

We have

$$y = \sqrt{d} \, Du, \qquad \ell = \ell(y), \qquad g_y := \nabla_y \ell, \qquad H_{yy} := \nabla_{yy}^2 \ell.$$

Remind two useful Jacobian matrices,

$$\frac{\partial u}{\partial z} = A = \frac{1}{s} P, \qquad \frac{\partial y}{\partial z} = \sqrt{d} \, DA = \frac{\sqrt{d}}{s} DP.$$

We will repeatedly use the chain rule through the path $W \to z \to u \to y \to \ell$, and similarly $\gamma \to D \to y \to \ell$.

## C.1. Part I: derivatives w.r.t. $\gamma$

**(1) First-order derivative.**  Since $y_i = \sqrt{d} \, \gamma_i u_i$, the Jacobian of $y$ w.r.t. $\gamma$ is diagonal:

$$\frac{\partial y}{\partial \gamma} = \sqrt{d} \, \text{Diag}(u).$$

Applying the chain rule $g_\gamma = \left(\frac{\partial y}{\partial \gamma}\right)^\top g_y$, we obtain

$$g_\gamma = \nabla_\gamma \ell = \sqrt{d} \, \text{Diag}(u) \, g_y = \sqrt{d} \, (u \odot g_y). \tag{13}$$

**(2) Second-order derivative.**  Because $y$ is linear in $\gamma$, $\frac{\partial^2 y}{\partial \gamma^2} = 0$, hence the Hessian comes purely from $H_{yy}$:

$$H_{\gamma\gamma} = \nabla_{\gamma\gamma}^2 \ell = \left(\frac{\partial y}{\partial \gamma}\right)^\top H_{yy} \left(\frac{\partial y}{\partial \gamma}\right) = d \, \text{Diag}(u) \, H_{yy} \, \text{Diag}(u). \tag{14}$$

## C.2. Part II: derivatives w.r.t. $W$

**(1) First-order derivative.**  Step 1 ($y \to z$): gradient w.r.t. $z$

By the chain rule,

$$g_z = \left(\frac{\partial y}{\partial z}\right)^\top g_y = \left(\frac{\sqrt{d}}{s} DP\right)^\top g_y.$$

Using $P^\top = P$ and $D^\top = D$, this simplifies to

$$g_z = \frac{\sqrt{d}}{s} \, P D \, g_y.$$

Step 2 ($z = Wx \to W$): gradient w.r.t. $W$

We write the differential $dz = dW \, x$, so

$$d\ell = g_z^\top dz = g_z^\top (dW \, x) = \langle g_z x^\top, \, dW \rangle_F.$$

Matching coefficients under the Frobenius inner product yields

$$g_W = \nabla_W \ell = g_z x^\top = \frac{\sqrt{d}}{s} \, (PDg_y) \, x^\top. \tag{15}$$

If one needs the vectorized mapping,

$$\text{vec}(z) = \text{vec}(Wx) = (x^\top \otimes I) \, \text{vec}(W).$$

**(2) Second-order derivative.** We first derive $\boldsymbol{H_{zz}} := \nabla^2_{zz}\ell$, then lift it to $\mathrm{vec}(\boldsymbol{W})$.

Step 1: Hessian w.r.t. $\boldsymbol{z}$

The second-order chain rule gives

$$\boldsymbol{H_{zz}} = \left(\frac{\partial \boldsymbol{y}}{\partial \boldsymbol{z}}\right)^\top \boldsymbol{H_{yy}}\left(\frac{\partial \boldsymbol{y}}{\partial \boldsymbol{z}}\right) + \sum_{k=1}^{d}(\boldsymbol{g_y})_k\,\nabla^2_{zz}y_k.$$

The first term is the Gauss–Newton part:

$$\left(\frac{\partial \boldsymbol{y}}{\partial \boldsymbol{z}}\right)^\top \boldsymbol{H_{yy}}\left(\frac{\partial \boldsymbol{y}}{\partial \boldsymbol{z}}\right) = \frac{d}{s^2}\,\boldsymbol{P}\,\boldsymbol{D}\,\boldsymbol{H_{yy}}\,\boldsymbol{D}\,\boldsymbol{P}.$$

For the second term, we use the bilinear form of the normalization Hessian:

$$\nabla^2 \boldsymbol{u}(\boldsymbol{z})[\boldsymbol{a},\boldsymbol{b}] = -\frac{1}{s^2}\Big((\boldsymbol{u}^\top \boldsymbol{a})\,\boldsymbol{Pb} + (\boldsymbol{u}^\top \boldsymbol{b})\,\boldsymbol{Pa} + (\boldsymbol{a}^\top \boldsymbol{Pb})\,\boldsymbol{u}\Big).$$

Since $\boldsymbol{y} = \sqrt{d}\boldsymbol{Du}$,

$$\sum_{k=1}^{d}(\boldsymbol{g_y})_k\,\nabla^2 y_k[\boldsymbol{a},\boldsymbol{b}] = \boldsymbol{g_y}^\top \nabla^2 \boldsymbol{y}[\boldsymbol{a},\boldsymbol{b}] = \sqrt{d}\,(\boldsymbol{Dg_y})^\top \nabla^2 \boldsymbol{u}[\boldsymbol{a},\boldsymbol{b}].$$

Substituting gives

$$\sum_{k=1}^{d}(\boldsymbol{g_y})_k\,\nabla^2 y_k[\boldsymbol{a},\boldsymbol{b}] = -\frac{\sqrt{d}}{s^2}\Big((\boldsymbol{u}^\top \boldsymbol{a})(\boldsymbol{Dg_y})^\top \boldsymbol{Pb} + (\boldsymbol{u}^\top \boldsymbol{b})(\boldsymbol{Dg_y})^\top \boldsymbol{Pa} + (\boldsymbol{u}^\top \boldsymbol{Dg_y})\,\boldsymbol{a}^\top \boldsymbol{Pb}\Big).$$

Equivalently, the associated matrix form is

$$\sum_{k=1}^{d}(\boldsymbol{g_y})_k\,\nabla^2_{zz}y_k = -\frac{\sqrt{d}}{s^2}\Big(\boldsymbol{PDg_y}\boldsymbol{u}^\top + \boldsymbol{u}^\top \boldsymbol{Dg_y}\boldsymbol{P} + \boldsymbol{u}\boldsymbol{g_y}^\top \boldsymbol{DP}\Big).$$

Combining both terms,

$$\boldsymbol{H_{zz}} = \frac{d}{s^2}\,\boldsymbol{P}\,\boldsymbol{D}\,\boldsymbol{H_{yy}}\,\boldsymbol{D}\,\boldsymbol{P} - \frac{\sqrt{d}}{s^2}\Big(\boldsymbol{PDg_y}\boldsymbol{u}^\top + \boldsymbol{u}^\top \boldsymbol{Dg_y}\boldsymbol{P} + \boldsymbol{u}\boldsymbol{g_y}^\top \boldsymbol{DP}\Big), \qquad s = \|\boldsymbol{Wx}\|_2.$$

Using $\mathrm{vec}(\boldsymbol{z}) = (\boldsymbol{x}^\top \otimes \boldsymbol{I})\,\mathrm{vec}(\boldsymbol{W})$,

$$\begin{aligned}
\boldsymbol{H}_{\mathrm{vec}(\boldsymbol{W})\,\mathrm{vec}(\boldsymbol{W})} &= (\boldsymbol{xx}^\top) \otimes \boldsymbol{H_{zz}} \\
&= (\boldsymbol{xx}^\top) \otimes \left[\frac{d}{s^2}\,\boldsymbol{P}\,\boldsymbol{D}\,\boldsymbol{H_{yy}}\,\boldsymbol{D}\,\boldsymbol{P} - \frac{\sqrt{d}}{s^2}\Big(\boldsymbol{PDg_y}\boldsymbol{u}^\top + \boldsymbol{u}^\top \boldsymbol{Dg_y}\boldsymbol{P} + \boldsymbol{u}\boldsymbol{g_y}^\top \boldsymbol{DP}\Big)\right].
\end{aligned} \tag{16}$$

# D. Proof of Theorem 4.1

*Proof.* Recall SimpleNorm:

$$\boldsymbol{y} = \boldsymbol{\gamma} \odot \sqrt{d} \frac{\boldsymbol{W}\boldsymbol{x}}{\|\boldsymbol{W}\boldsymbol{x}\|_2}, \qquad \boldsymbol{z} = \boldsymbol{W}\boldsymbol{x}, \qquad s := \|\boldsymbol{z}\|_2, \qquad \boldsymbol{u} := \frac{\boldsymbol{z}}{s}, \qquad \boldsymbol{P} := \boldsymbol{I} - \boldsymbol{u}\boldsymbol{u}^\top.$$

Assume $\boldsymbol{D} = \mathrm{Diag}(\boldsymbol{\gamma}) = \boldsymbol{I}$. Let $\ell = \ell(\boldsymbol{y})$ and define

$$\boldsymbol{g_y} := \nabla_{\boldsymbol{y}}\ell(\boldsymbol{y}) \in \mathbb{R}^d, \qquad \boldsymbol{H_{yy}} := \nabla^2_{\boldsymbol{yy}}\ell(\boldsymbol{y}) \in \mathbb{R}^{d\times d}.$$

By the chain rule, we have the decomposition:

$$\boldsymbol{H_{xx}} = \nabla^2_{\boldsymbol{x}}\ell = \underbrace{\boldsymbol{J_x^y}^\top \boldsymbol{H_{yy}} \boldsymbol{J_x^y}}_{\text{Gauss–Newton term}} + \underbrace{\boldsymbol{C}}_{\text{curvature term}} \tag{17}$$

Here the Jacobian is

$$\boldsymbol{J_x^y} = \frac{\sqrt{d}}{s} \boldsymbol{D} \boldsymbol{P} \boldsymbol{W} = \frac{\sqrt{d}}{s} \boldsymbol{P} \boldsymbol{W},$$

and therefore the Gauss–Newton term is

$$\boldsymbol{L} = (\boldsymbol{J_x^y})^\top \boldsymbol{H_{yy}} \boldsymbol{J_x^y} = \frac{d}{s^2} \boldsymbol{W}^\top \boldsymbol{P} \boldsymbol{H_{yy}} \boldsymbol{P} \boldsymbol{W}. \tag{18}$$

The curvature term (with the condition $\boldsymbol{D} = \boldsymbol{I}$) can be written as

$$\boldsymbol{C} = -\frac{\sqrt{d}}{s^2} \boldsymbol{W}^\top \Big( (\boldsymbol{P}\boldsymbol{g_y})\boldsymbol{u}^\top + (\boldsymbol{u}^\top \boldsymbol{g_y})\boldsymbol{P} + \boldsymbol{u}(\boldsymbol{g_y}^\top \boldsymbol{P}) \Big) \boldsymbol{W}, \tag{19}$$

where $\boldsymbol{u}^\top \boldsymbol{g_y}$ is a scalar.

**Bounding the Gauss–Newton term.** Define

$$\kappa := \frac{\sqrt{d}}{\|\boldsymbol{W}\boldsymbol{x}\|_2} \|\boldsymbol{W}\|_2$$

We show that $\kappa = \Theta(1)$ under high effective rank and typical input. Let $\boldsymbol{x} = \sqrt{d}\,\xi$ where $\xi$ is isotropic (e.g., uniform on the sphere or subgaussian). Then

$$\mathbb{E}\|\boldsymbol{W}\boldsymbol{x}\|_2^2 = d\,\mathbb{E}\,\xi^\top \boldsymbol{W}^\top \boldsymbol{W} \xi = \|\boldsymbol{W}\|_F^2,$$

and by standard concentration for quadratic forms,

$$\|\boldsymbol{W}\boldsymbol{x}\|_2^2 \asymp \|\boldsymbol{W}\|_F^2 \quad \Longrightarrow \quad \|\boldsymbol{W}\boldsymbol{x}\|_2 \asymp \|\boldsymbol{W}\|_F \qquad \text{with high probability.}$$

Thus

$$\kappa = \frac{\sqrt{d}}{\|\boldsymbol{W}\boldsymbol{x}\|_2} \|\boldsymbol{W}\|_2 \asymp \sqrt{d}\, \frac{\|\boldsymbol{W}\|_2}{\|\boldsymbol{W}\|_F} = \sqrt{\frac{d}{r_{\text{eff}}(\boldsymbol{W})}}, \qquad r_{\text{eff}}(\boldsymbol{W}) := \frac{\|\boldsymbol{W}\|_F^2}{\|\boldsymbol{W}\|_2^2}.$$

If $r_{\text{eff}}(\boldsymbol{W}) \asymp cd$, then $\kappa \asymp 1/\sqrt{c} = \Theta(1)$ with high probability.

Additionally, define

$$\tau := \frac{\|\widetilde{\boldsymbol{W}}^\top \boldsymbol{P} \boldsymbol{H_{yy}} \boldsymbol{P} \widetilde{\boldsymbol{W}}\|_2}{\|\boldsymbol{H_{yy}}\|_2} \in [0,1], \qquad \widetilde{\boldsymbol{W}} := \frac{\boldsymbol{W}}{\|\boldsymbol{W}\|_2}.$$

Under the theorem's "non-pathological alignment" assumption (i.e., the dominant spectral modes of $\boldsymbol{H_{yy}}$ are not eliminated by projecting out $\mathrm{span}(\boldsymbol{u})$ and are represented in the range of $\boldsymbol{W}$, which has high effective rank), we have $\tau = \Theta(1)$.

Now, it follows that

$$\boldsymbol{L} = \frac{d}{\|\boldsymbol{W}\boldsymbol{x}\|_2^2} \boldsymbol{W}^\top \boldsymbol{P} \boldsymbol{H_{yy}} \boldsymbol{P} \boldsymbol{W} = \frac{\kappa^2}{\|\boldsymbol{W}\|_2^2} \boldsymbol{W}^\top \boldsymbol{P} \boldsymbol{H_{yy}} \boldsymbol{P} \boldsymbol{W} = \kappa^2 \widetilde{\boldsymbol{W}}^\top \boldsymbol{P} \boldsymbol{H_{yy}} \boldsymbol{P} \widetilde{\boldsymbol{W}}$$

and

$$\|\boldsymbol{L}\|_2 = \tau\,\kappa^2\,\|\boldsymbol{H_{yy}}\|_2 \qquad \text{where } \tau = \Theta(1) \text{ and } \kappa = \Theta(1) \quad [\text{w.h.p}] \tag{20}$$

**Bounding the curvature term.** From Equation 19 and submultiplicativity,

$$\|\boldsymbol{C}\|_2 \leq \frac{\sqrt{d}}{s^2} \|\boldsymbol{W}\|_2^2 \left\| (\boldsymbol{P}\boldsymbol{g_y})\boldsymbol{u}^\top + (\boldsymbol{u}^\top \boldsymbol{g_y})\boldsymbol{P} + \boldsymbol{u}(\boldsymbol{g_y}^\top \boldsymbol{P}) \right\|_2.$$

Now use $\|\boldsymbol{u}\|_2 = 1$, $\|\boldsymbol{P}\|_2 = 1$, and $\|\boldsymbol{P}\boldsymbol{g_y}\|_2 \leq \|\boldsymbol{g_y}\|_2$:

$$\|(\boldsymbol{P}\boldsymbol{g_y})\boldsymbol{u}^\top\|_2 = \|\boldsymbol{P}\boldsymbol{g_y}\|_2 \|\boldsymbol{u}\|_2 \leq \|\boldsymbol{g_y}\|_2,$$

$$\|(\boldsymbol{u}^\top \boldsymbol{g_y})\boldsymbol{P}\|_2 = |\boldsymbol{u}^\top \boldsymbol{g_y}| \, \|\boldsymbol{P}\|_2 \leq \|\boldsymbol{g_y}\|_2,$$

$$\|\boldsymbol{u}(\boldsymbol{g_y}^\top \boldsymbol{P})\|_2 = \|\boldsymbol{u}\|_2 \|\boldsymbol{P}\boldsymbol{g_y}\|_2 \leq \|\boldsymbol{g_y}\|_2.$$

By triangle inequality, the middle norm is at most $3\|\boldsymbol{g_y}\|_2$, hence

$$\|\boldsymbol{C}\|_2 \leq \frac{3\sqrt{d}}{\|\boldsymbol{W}\boldsymbol{x}\|_2^2} \|\boldsymbol{W}\|_2^2 \|\boldsymbol{g_y}\|_2 = \frac{3\kappa^2}{\sqrt{d}} \|\boldsymbol{g_y}\|_2. \tag{21}$$

**Dominance of $\boldsymbol{L}$ over $\boldsymbol{C}$.** Combining (20) and (21),

$$\frac{\|\boldsymbol{C}\|_2}{\|\boldsymbol{L}\|_2} \leq \frac{3}{\tau\sqrt{d}} \frac{\|\boldsymbol{g_y}\|_2}{\|\boldsymbol{H_{yy}}\|_2}.$$

Assuming $\tau = \Theta(1)$ (non-pathological alignment) and $\|\boldsymbol{g_y}\|_2/\|\boldsymbol{H_{yy}}\|_2 = O(1)$ (bounded gradient-to-curvature ratio, as stated in the theorem assumptions), we obtain

$$\frac{\|\boldsymbol{C}\|_2}{\|\boldsymbol{L}\|_2} = O(d^{-1/2}),$$

so in high dimension $\|\boldsymbol{L}\|_2 \gg \|\boldsymbol{C}\|_2$ with high probability. Therefore the Gauss–Newton term dominates $\boldsymbol{H_{xx}}$ in typical non-pathological regimes.

This completes the proof of Theorem 4.1. $\qquad\square$

# E. Proof of Theorem 4.2

*Proof.* Provided $\ell = \ell(\boldsymbol{y})$ is twice differentiable. Denote

$$\boldsymbol{g_y} := \nabla_{\boldsymbol{y}} \ell, \qquad \boldsymbol{H_{yy}} := \nabla^2_{\boldsymbol{yy}} \ell,$$

We have the two mappings:

Linear: $\quad \boldsymbol{y}_1 = \boldsymbol{W}_1 \boldsymbol{x}, \qquad \boldsymbol{H}^{\text{lin}}_{\boldsymbol{xx}} = \boldsymbol{W}_1^\top \boldsymbol{H}_{\boldsymbol{y}_1 \boldsymbol{y}_1} \boldsymbol{W}_1.$

SimpleNorm: $\quad \boldsymbol{y}_2 = \boldsymbol{D} \dfrac{\sqrt{d}\, \boldsymbol{W}_2 \boldsymbol{x}}{\|\boldsymbol{W}_2 \boldsymbol{x}\|_2}$, where $\boldsymbol{D} = \text{Diag}(\boldsymbol{\gamma})$ and define

$$\boldsymbol{z} = \boldsymbol{W}_2 \boldsymbol{x}, \quad s = \|\boldsymbol{z}\|_2, \quad \boldsymbol{u} = \boldsymbol{z}/s, \quad \boldsymbol{P} = \boldsymbol{I} - \boldsymbol{u}\boldsymbol{u}^\top.$$

The Hessian w.r.t. $\boldsymbol{x}$ admits the standard decomposition

$$\boldsymbol{H}^{\text{sn}}_{\boldsymbol{xx}} = \boldsymbol{L} + \boldsymbol{C}, \qquad \boldsymbol{L} = \boldsymbol{J}_{\boldsymbol{x}}^{\boldsymbol{y}_2 \top} \boldsymbol{H}_{\boldsymbol{y}_2 \boldsymbol{y}_2} \boldsymbol{J}_{\boldsymbol{x}}^{\boldsymbol{y}_2},$$

where $\boldsymbol{J}_{\boldsymbol{x}}^{\boldsymbol{y}_2}$ is the Jacobian of $\boldsymbol{y}_2$ w.r.t. $\boldsymbol{x}$, and $\boldsymbol{C}$ is the curvature term induced by the normalization.

Assuming the high-dimensional conditions stated in Theorem 4.1 hold ($\|\boldsymbol{x}\|_2 = \sqrt{d}$, $\boldsymbol{D} = \boldsymbol{I}$, $\|\boldsymbol{P}\|_2 = 1$, $\boldsymbol{W}_2$ has high effective rank, no pathological alignment, and $\|\boldsymbol{g_y}\|_2/\|\boldsymbol{H_{yy}}\|_2 = O(1)$), Theorem 4.1 shows that the Gauss-Newton term dominates the curvature term, i.e., $\|\boldsymbol{L}\|_2 \gg \|\boldsymbol{C}\|_2$. Therefore, we obtain the approximation $\|\boldsymbol{H}^{\text{sn}}_{\boldsymbol{xx}}\|_2 \approx \|\boldsymbol{L}\|_2$.

We now show an integral result: the spectral norm of $\boldsymbol{H}^{\text{lin}}_{\boldsymbol{xx}}$ is directly proportional to the spectral norm of the weight matrix $\boldsymbol{W}$ whereas the spectral norm of $\boldsymbol{H}^{\text{sn}}_{\boldsymbol{xx}}$ is independent of the spectral norm of the weight matrix. In the following, we assume $\boldsymbol{W} = \boldsymbol{W}_1 = \boldsymbol{W}_2$ and $\boldsymbol{H}_{\boldsymbol{y}_1 \boldsymbol{y}_1} = \boldsymbol{H}_{\boldsymbol{y}_2 \boldsymbol{y}_2} := \boldsymbol{H_{yy}}$.

Define $\widetilde{\boldsymbol{W}} := \dfrac{\boldsymbol{W}}{\alpha}$ where $\alpha := \|\boldsymbol{W}\|_2$. Consequently, $\|\widetilde{\boldsymbol{W}}\|_2 = 1$ and $\boldsymbol{W} = \alpha \widetilde{\boldsymbol{W}}$. Now, treating $\boldsymbol{H}^{\text{lin}}_{\boldsymbol{xx}}$ and $\boldsymbol{H}^{\text{sn}}_{\boldsymbol{xx}}$ as functions of $\alpha$, it follows that:

$$\|\boldsymbol{H}^{\text{lin}}_{\boldsymbol{xx}}(\alpha)\|_2 = \|\boldsymbol{W}_1^\top \boldsymbol{H}_{\boldsymbol{y}_1 \boldsymbol{y}_1} \boldsymbol{W}_1\|_2 = \alpha^2 \|\widetilde{\boldsymbol{W}}^\top \boldsymbol{H_{yy}} \widetilde{\boldsymbol{W}}\|_2 \tag{22}$$

and

$$\|\boldsymbol{H}^{\text{sn}}_{\boldsymbol{xx}}(\alpha)\|_2 \approx \|\boldsymbol{L}(\alpha)\|_2 = \frac{d}{\|\boldsymbol{W}\boldsymbol{x}\|_2^2} \|\boldsymbol{W}^\top \boldsymbol{P} \boldsymbol{D} \boldsymbol{H_{yy}} \boldsymbol{D} \boldsymbol{P} \boldsymbol{W}\|_2 = \frac{d}{\|\widetilde{\boldsymbol{W}}\boldsymbol{x}\|_2^2} \|\widetilde{\boldsymbol{W}}^\top \boldsymbol{P} \boldsymbol{H_{yy}} \boldsymbol{P} \widetilde{\boldsymbol{W}}\|_2 \tag{23}$$

Note that $\boldsymbol{P} = \boldsymbol{I} - \boldsymbol{u}\boldsymbol{u}^\top$ is independent of $\alpha$:

$$\boldsymbol{u}(\alpha) = \frac{\boldsymbol{W}\boldsymbol{x}}{\|\boldsymbol{W}\boldsymbol{x}\|_2} = \frac{\alpha \widetilde{\boldsymbol{W}}\boldsymbol{x}}{\|\alpha \widetilde{\boldsymbol{W}}\boldsymbol{x}\|_2} = \frac{\widetilde{\boldsymbol{W}}\boldsymbol{x}}{\|\widetilde{\boldsymbol{W}}\boldsymbol{x}\|_2}, \qquad \Longrightarrow \qquad \boldsymbol{P}(\alpha) = \boldsymbol{I} - \boldsymbol{u}(\alpha)\boldsymbol{u}(\alpha)^\top = \boldsymbol{P}(1).$$

Therefore, $\|\boldsymbol{H}^{\text{lin}}_{\boldsymbol{xx}}(\alpha)\|_2$ depends quadratically on $\alpha$ whereas $\|\boldsymbol{H}^{\text{sn}}_{\boldsymbol{xx}}(\alpha)\|_2$ is *scale-invariant* in $\alpha$.

Next, we compare the magnitudes of the two Hessians. First, recall the definition of $\kappa$ provided in Theorem 4.1:

$$\kappa := \frac{\sqrt{d}}{\|\boldsymbol{W}\boldsymbol{x}\|_2} \|\boldsymbol{W}\|_2 = \frac{\sqrt{d}}{\alpha \|\widetilde{\boldsymbol{W}}\boldsymbol{x}\|_2} \alpha \|\widetilde{\boldsymbol{W}}\|_2 = \frac{\sqrt{d}}{\|\widetilde{\boldsymbol{W}}\boldsymbol{x}\|_2}$$

By definition, $\kappa^2 = \dfrac{d}{\|\widetilde{\boldsymbol{W}}\boldsymbol{x}\|_2^2}$, and Theorem 4.1 implies $\kappa^2 = \Theta(1)$ w.h.p.

Moreover, since $\|\widetilde{\boldsymbol{W}}\|_2 = \|\boldsymbol{P}\|_2 = 1$,

$$\|\boldsymbol{H}^{\text{sn}}_{\boldsymbol{xx}}(\alpha)\|_2 \approx \|\boldsymbol{L}\|_2 = \kappa^2 \|\widetilde{\boldsymbol{W}}^\top \boldsymbol{P} \boldsymbol{H_{yy}} \boldsymbol{P} \widetilde{\boldsymbol{W}}\|_2 \leq \kappa^2 \|\boldsymbol{H_{yy}}\|_2 \tag{24}$$

For the linear module, there exists a constant $c_{\text{lin}} > 0$ such that

$$\|\boldsymbol{H}_{\boldsymbol{xx}}^{\mathrm{lin}}(\alpha)\|_2 = \alpha^2 \|\widetilde{\boldsymbol{W}}^\top \boldsymbol{H}_{\boldsymbol{yy}} \widetilde{\boldsymbol{W}}\|_2 \geq \alpha^2 c_{\mathrm{lin}} \|\boldsymbol{H}_{\boldsymbol{yy}}\|_2.$$

We assume $\mathrm{Range}(\widetilde{\boldsymbol{W}})$ is not adversarially aligned with the leading eigenspace of $\boldsymbol{H}_{\boldsymbol{yy}}$; in particular the overlap is constant-order, so the visibility constant satisfies $c_{\mathrm{lin}} = \Theta(1)$ and does not decay with $d$.

Combining all the aforementioned results, the final ratio between the two Hessians satisfies:

$$\frac{\|\boldsymbol{H}_{\boldsymbol{xx}}^{\mathrm{lin}}(\alpha)\|_2}{\|\boldsymbol{H}_{\boldsymbol{xx}}^{\mathrm{sn}}(\alpha)\|_2} \geq \frac{\alpha^2 \|\widetilde{\boldsymbol{W}}^\top \boldsymbol{H}_{\boldsymbol{yy}} \widetilde{\boldsymbol{W}}\|_2}{\kappa^2 \|\boldsymbol{H}_{\boldsymbol{yy}}\|_2} \geq \alpha^2 \frac{c_{\mathrm{lin}}}{\kappa^2}$$

Empirically, $\alpha = \|\boldsymbol{W}\|_2$ typically grows to tens or hundreds during training, so $\alpha^2 c_{\mathrm{lin}} \gg \kappa^2$ (recall $\kappa = \Theta(1)$ w.h.p.). Hence:

$$\|\boldsymbol{H}_{\boldsymbol{xx}}^{\mathrm{lin}}\|_2 \gg \|\boldsymbol{H}_{\boldsymbol{xx}}^{\mathrm{sn}}\|_2 \qquad \text{(with high probability)}.$$

In summary, in non-pathological cases, the gradient Lipschitz constant of $\|\boldsymbol{H}_{\boldsymbol{xx}}^{\mathrm{lin}}\|_2$ is much larger than the gradient Lipschitz constant of $\|\boldsymbol{H}_{\boldsymbol{xx}}^{\mathrm{sn}}\|_2$.

This completes the proof of Theorem 4.2.

$\square$

# F. Empirical Support for Learned RMSNorm Gain Parameter Assumption

Theorem 4.1 is stated under the simplifying assumption $D = \mathrm{Diag}(\gamma) = I$. This assumption is merely used to make the main curvature mechanism transparent, and should not be interpreted as requiring the learned RMSNorm gain parameters to be exactly equal to one in practice. What is needed for the scale-control argument is that the entries of $\gamma$ remain bounded by a moderate constant, so that the gain parameters do not dominate the normalization effect.

In practice, this is generally upheld, as shown in Figure 9, where we record the learned RMSNorm gain parameters across all layers for Llama2-7B, Llama2-7B with QKNorm, and SimpleGPT-7B.

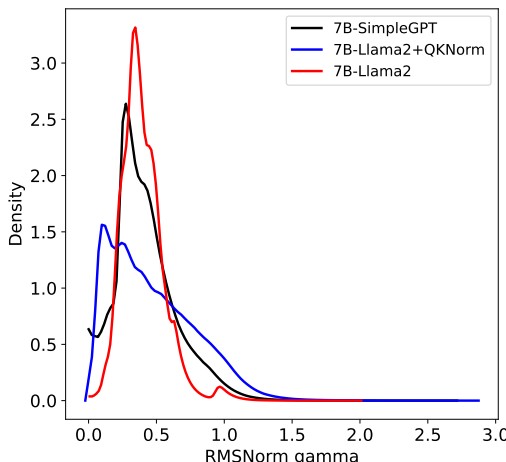

*Figure 9.* Distribution of learned RMSNorm gain parameters $\gamma$ across Llama2-7B, Llama2-7B with QKNorm, and SimpleGPT-7B.

# G. Detailed Experimental Settings

Our **SimpleGPT** models are built on Llama2, Llama3, and nanoGPT (a GPT-2 implementation). We apply the SimpleNorm operator to all Transformer blocks except the embedding layer and classification layer. Our implementations are based on the Adam-mini [1] (Zhang et al., 2024) and nanoGPT [2] (Karpathy, 2022). Below we briefly describe the main architectural and training settings for each backbone.

**nanoGPT** is a lightweight and efficient implementation of the GPT-2 architecture. It uses the GELU activation function and a byte-pair encoding (BPE) tokenizer (Gage, 1994) consistent with GPT-2 (Radford et al., 2019), with an expanded vocabulary size of 50,257 tokens. 2,000 steps are used for learning rate warmup. Our training data on nanoGPT models is OpenWebText.

**Llama2** adopts the SwiGLU (Shazeer, 2020) activation function in the feed-forward networks, which improves expressivity and parameter efficiency. Positional information is encoded using Rotary Positional Embeddings (RoPE) (Su et al., 2023). Llama2 also introduces Grouped-Query Attention (GQA) to reduce inference-time memory usage and computational cost. The model uses a SentencePiece-based BPE tokenizer with a vocabulary size of 32K tokens. In our experiments, 1% of the total steps are allocated for learning rate warmup. Our training data on Llama2 models is C4.

**Llama3** follows the dense Transformer design of Llama2, while introducing several targeted changes. It continues to use GQA with eight key-value heads to improve decoding efficiency and reduce key-value cache size. A major difference lies in the tokenizer: Llama3 adopts a significantly larger vocabulary of 128K tokens, combining tokens from the tiktoken tokenizer with additional multilingual tokens, which improves compression rates and language coverage. To better support long contexts, the RoPE base frequency is increased to 500,000. In our experiments, 1% of the total steps are allocated for learning rate warmup. Our training data on Llama3 models is C4.

Across all experiments, we adopt the AdamW optimizer (Kingma & Ba, 2014; Loshchilov & Hutter, 2019) with $\beta_1 = 0.9$ and $\beta_2 = 0.95$. Since we know that weight decay is associated with the learning rate, and our method permits the use of larger learning rates, we accordingly adjust the weight decay. Unless otherwise stated, a weight decay value of 0.1 is used throughout our experiments. Additional hyperparameter configurations are summarized in Table 2.

All models are trained using PyTorch (Paszke et al., 2019) with bfloat16 precision on A800 GPUs. We employ a cosine learning rate schedule for all training runs.

# H. Parameters and configurations of SimpleGPT

*Table 2.* Model configurations for different scales of SimpleGPT. The 1B and 7B models are based on Llama2, the 8B model is based on Llama3, and the 1.4B model is based on nanoGPT.

| | SimpleGPT-**1B** | SimpleGPT-**7B** | SimpleGPT-**8B** | SimpleGPT-**1.4B** |
|---|---|---|---|---|
| Origin from | Llama2 | Llama2 | Llama3 | nanoGPT(GPT2) |
| Layers | 18 | 32 | 32 | 48 |
| Model Dimension | 2,048 | 4,096 | 4,096 | 1,536 |
| FFN Dimension | 5,632 | 11,008 | 14,336 | 6,144 |
| Attention Heads | 16 | 32 | 32 | 24 |
| Key / Value Heads | 16 | 32 | 8 | 24 |
| Activation Function | SwiGLU | SwiGLU | SwiGLU | GeLU |
| Vocabulary Size | 32,000 | 32,000 | 128,000 | 50,304 |
| Positional Embeddings (RoPE) | $\theta = 10,000$ | $\theta = 10,000$ | $\theta = 500,000$ | No |
| Batch Size | $512 \times 256$ | $2048 \times 192$ | $2048 \times 192$ | $1024 \times 512$ |
| Training Steps | 200K | 20K/40K/60K | 20K | 100K |
| Warmup Steps | 1% | 1% | 1% | 2000 |

---

[1] https://github.com/zyushun/Adam-mini
[2] https://github.com/karpathy/nanoGPT

# I. Comparison with SandwichNorm

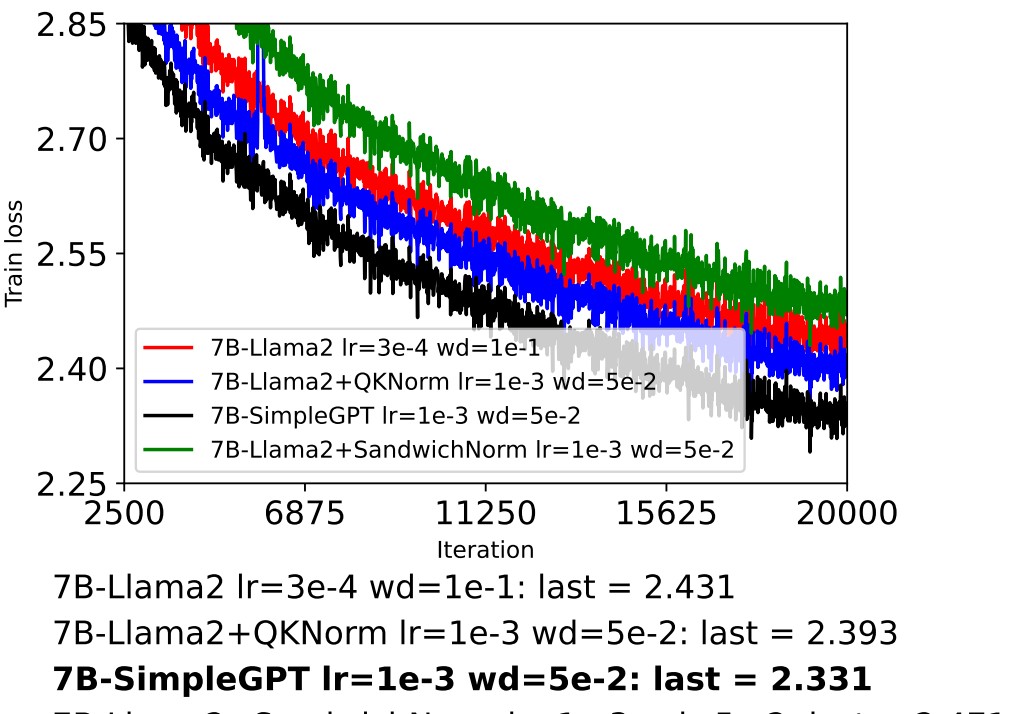

*Figure 10.* Comparison among Llama2-7B, Llama2-7B with QKNorm, Llama2-7B with SandwichNorm, and SimpleGPT-7B.

To examine whether the gains of SimpleGPT come merely from adding more normalization layers, we compare against SandwichNorm (Ding et al., 2021), which inserts normalization around Transformer sublayers. As shown in Figure 10, SimpleGPT-7B achieves lower training loss than Llama2-7B with SandwichNorm. This suggests that the improvement is not simply due to increasing the number of normalization operations, but is more closely related to the SimpleNorm placement principle of normalizing immediately after linear mappings.

## J. More experiments on SimpleGPT-7B

Furthermore, we evaluate the SimpleGPT-7B models using different learning rates or weight decay values. Results are shown in Figure 11 and Figure 12.

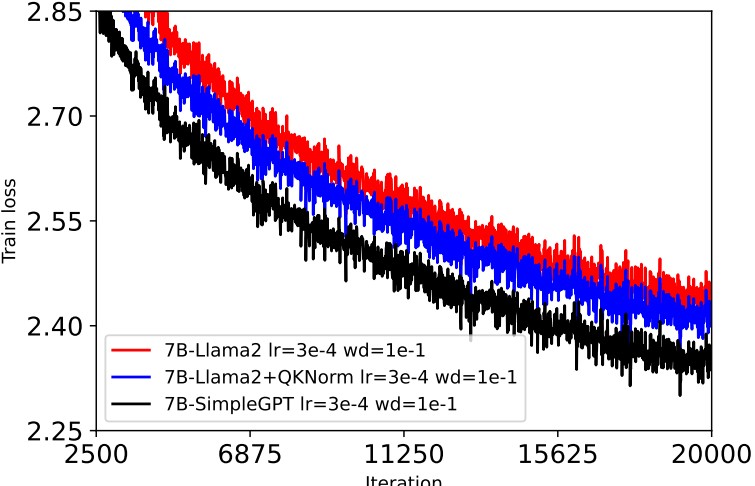

7B-Llama2 lr=3e-4 wd=1e-1: last = 2.431

7B-Llama2+QKNorm lr=3e-4 wd=1e-1: last = 2.404

**7B-SimpleGPT lr=3e-4 wd=1e-1: last = 2.340**

*Figure 11.* The training loss curves of Llama2-7B, Llama2-7B with QKNorm and SimpleGPT-7B with learning rate 3e-4 and weight decay 0.1.

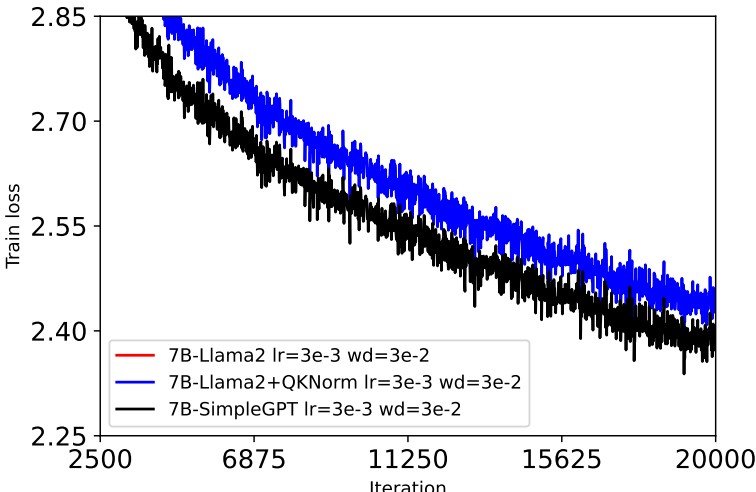

7B-Llama2 lr=3e-3 wd=3e-2: last = 4.579

7B-Llama2+QKNorm lr=3e-3 wd=3e-2: last = 2.430

**7B-SimpleGPT lr=3e-3 wd=3e-2: last = 2.377**

*Figure 12.* The training loss curves of Llama2-7B, Llama2-7B with QKNorm and SimpleGPT-7B with learning rate 3e-3 and weight decay 0.03.

## K. More experiments on SimpleGPT-8B

Furthermore, we evaluate the SimpleGPT-8B models using different learning rates or weight decay values. Results are shown in Figure 13 and Figure 14.

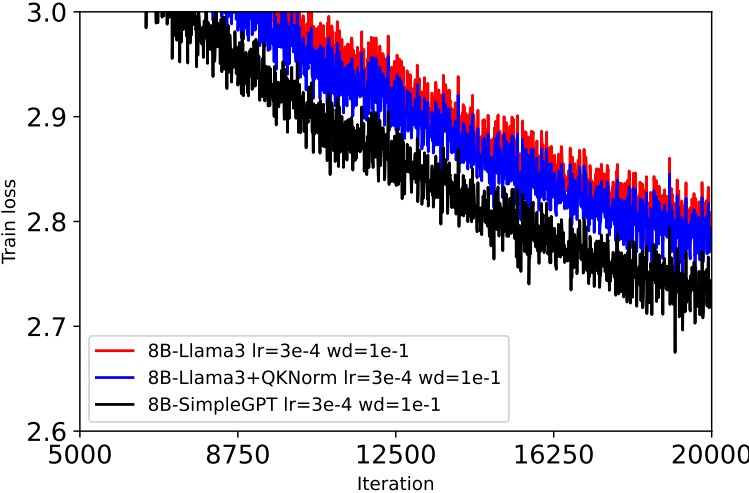

8B-Llama3 lr=3e-4 wd=1e-1: last = 2.779

8B-Llama3+QKNorm lr=3e-4 wd=1e-1: last = 2.763

**8B-SimpleGPT lr=3e-4 wd=1e-1: last = 2.709**

*Figure 13.* The training loss curves of Llama3-8B, Llama3-8B with QKNorm and SimpleGPT-8B with learning rate 3e-4 and weight decay 0.1.

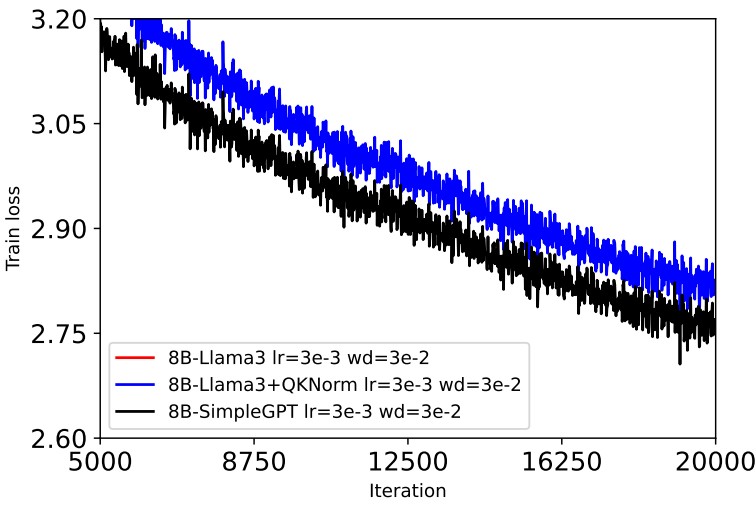

8B-Llama3 lr=3e-3 wd=3e-2: last = 5.255

8B-Llama3+QKNorm lr=3e-3 wd=3e-2: last = 2.800

**8B-SimpleGPT lr=3e-3 wd=3e-2: last = 2.738**

*Figure 14.* The training loss curves of Llama3-8B, Llama3-8B with QKNorm and SimpleGPT-8B with learning rate 3e-3 and weight decay 0.03.

# L. More experiments on weight decays

We conduct experiments on the SimpleGPT-8B model using two different weight decay values to evaluate robustness to regularization. Across all tested settings, SimpleGPT-8B consistently outperforms Llama2-8B with QKNorm. These results indicate that the benefits of SimpleNorm are not sensitive to the choice of weight decay, further demonstrating its robustness in large-scale training. Results are shown in Figure 15.

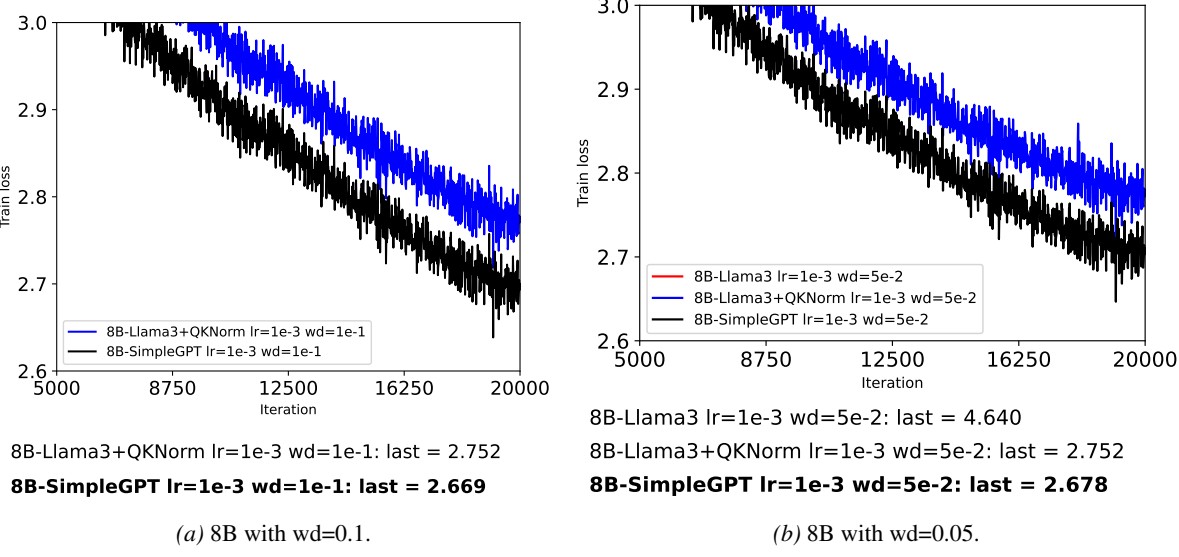

8B-Llama3+QKNorm lr=1e-3 wd=1e-1: last = 2.752

**8B-SimpleGPT lr=1e-3 wd=1e-1: last = 2.669**

*(a)* 8B with wd=0.1.

8B-Llama3 lr=1e-3 wd=5e-2: last = 4.640

8B-Llama3+QKNorm lr=1e-3 wd=5e-2: last = 2.752

**8B-SimpleGPT lr=1e-3 wd=5e-2: last = 2.678**

*(b)* 8B with wd=0.05.

*Figure 15.* Overall comparison across Llama3-8B with QKNorm and SimpleGPT-1B under two different weight decay values.

