# OpenReview forum: "SimpleGPT: Improving GPT via A Simple Normalization Strategy"
_ICML.cc/2026/Conference — ICML 2026 regular_

### Official Review · Reviewer_iFri · 2026-03-02

**Soundness:** 2
**Presentation:** 2
**Significance:** 2
**Originality:** 3
**Overall Recommendation:** 4
**Confidence:** 4

**Summary:**

This paper introduces a new normalization strategy, which allows the use of larger learning rates when training large language models.
Unlike previous works that only provides empirical justification, the authors connect their design to theorectial stability conditions.
Their mathematical analysis shows that their strategy can reduce the spectral norm of the Hessian matrix and thus tolerate large
learning rates. Experimental results over diﬀerent models further demonstrate the eﬀectiveness of their normalization.

**Compliance With Llm Reviewing Policy:**

Affirmed.

**Final Justification:**

I raise my decision to weakly accept.

The rebuttal supports the evidence to our concerned question.

**Key Questions For Authors:**

Q1: With respect to Weakness 1, could you provide additional evidence to justify the validity of the assumption and the corresponding
theoretical interpretation? For example, reporting the empirical range of the gain parameters would help assess whether the
assumption is satisfied in practice.

Q2: Could you explain the connections between the derivations of $\gamma$, $W$ and Theorem 4.2 in the main body?

Q3: As there are a lot normalization schemes reviewed in Section 2, why do you select QKNorm for comparison?

**Limitations:**

The authors have not discussed the limitations of their work. The experiment has room for improvement, as noted in the third
weakness mentioned above.

**Strengths And Weaknesses:**

Strengths:
1. A notable strength of the paper is its attempt to ground the proposed normalization strategy in theoretical stability analysis.
The authors provide a mathematical argument linking their method to Hessian spectral norm reduction and learning rate
stability.
2. The extensive evaluation across diverse architectures, including GPT-2, LLaMA-2, and LLaMA-3, demonstartes the partical
eﬀectivenss of the proposed strategy. Their strategy appears to be promising and worth further exploration.

Weakness:
1. However, their analysis relies on an assumption that may not hold in practice, especially when the proposed normalization strategy is applied. The assumption is
     $$
     D = \text{Diag}(\mathbf{\gamma}) = I,
     $$

     where $\gamma \in \mathbb{R}^d$ is the gain paramter, and $I$ is a $d$-by-$d$ identity matrix. This assumption indicates that all gain paramters are ones, which is unlikely to be ture. Note that all linear projections in an LLM are replaced with the proposed SimpleNorm operation,
     $$
     \gamma \odot \sqrt{d} \frac{W\mathbb{x}}{\Vert W\mathbb{x} \Vert_2},
     $$

     where $\mathbb{x} \in \mathbb{R}^m$ is the input embedding of a linear projection, and $W \in \mathbb{R}^{d \times m}$ is the weight matrix of the linear projection. In this operation, the gain paramter $\gamma$ is the only factor controlling the output scale, and thus it is difficult to justify the assumption that all gain parameters are equal to one in every linear projection of an LLM. Therefore, the current theoretical analysis may not be sufficient to fully explain the reported empirical improvements. The two main theorems (Theorems 4.1 and 4.2) in this paper are based on this assumption, which does not align with real-world conditions, seriously impacting the significance and soundness of the paper.

2. The writing in this work is not very good. The symbols are not properly defined when they first appear, such as the variables $\mathbb{x}$ and $\mathbb{y}$ on the right side of Line 21, and some notations are abused, like $\mathbb{y}$. Additionally, the connections between the derivations of $\gamma, W$ and Theorem 4.2 are not clearly explained in the paper.

3. The experiments are not rigorous enough. Rather than their proposed strategy, the fact that a larger learning rate could be used in the experiments can be possibly because the authors applied normalization to all linear projections. In light of the first weakness, the authors need to conduct more ablation studies to validate their theoretical analysis. For example, they could add RMSNorm before all linear projections, train a model, and include it in the experiments for comparison. In addition, the reason behind the choice of QKNorm for comparison is unclear.

---

> ### Author Rebuttal · Authors · 2026-03-31
>
> We sincerely thank the reviewer for the constructive suggestions.  _In this paper, we provide a novel perspective for understanding and designing Transformer models from the lens of second-order geometry._
>
> To address your concerns, we added further analysis and experiments.
> We hope our responses address your concerns.
>
> **Q1: on the assumptions**
> > However, their analysis relies on an assumption that may not hold in practice, especially when the proposed normalization strategy is applied. The assumption is $$D = \text{Diag}(\mathbf{\gamma}) = I.$$.......The two main theorems (Theorems 4.1 and 4.2) in this paper are based on this assumption, which does not align with real-world conditions, seriously impacting the significance and soundness of the paper.
>
> > Q1: With respect to Weakness 1, ..., reporting the empirical range of the gain parameters would help assess whether the assumption is satisfied in practice.
>
> **Response:** Thank you very much for the suggestion. To address this concern, we analyzed the distribution of the RMSNorm scaling parameter $\mathbf{\gamma}$ across all layers in our three 7B models: LLaMA2-7B with PreNorm, LLaMA2-7B with PreNorm+QKNorm, and SimpleGPT-7B. The results are shown in Figure 3 of the anonymous link included at the end. We found that in all three models, $\mathbf{\gamma}$ lies in a relatively fixed range between 0 and 1.5. This suggests that in typical large language models, $\mathbf{\gamma}$ does not become excessively large during training.
>
> In addition, our theory only requires $\mathbf{\gamma}$ to be $O(1)$. That is, as long as its magnitude does not grow far beyond a moderate constant (e.g., far beyond 10), the analysis remains valid. Such assumptions are common in mathematical proofs, and here they are also supported by empirical observations. Therefore, the assumption on $\mathbf{\gamma}$ is mild and realistic.
>
> ---
>
> **Q2: on the writing**
> > The writing in this work is not very good. The symbols are not properly defined,.... Additionally, the connections between the derivations of $\gamma, W$ and Theorem 4.2 are not clearly explained in the paper.
>
> **Response:** Thank you for the comment. We will revise the paper to define symbols more carefully and make the logical flow clearer.
>
> In the current chain-rule derivations, our main focus is on derivatives with respect to the activations $\mathbf{x}^l$, since they are most directly related to gradient propagation across layers. By contrast, in optimization in optimizers, we are often more concerned with derivatives with respect to the parameters $\mathbf{\gamma}$ and $\mathbf{W}$.
>
> Our derivations for $\mathbf{\gamma}$ and $\mathbf{W}$ are included mainly to provide intuition for how the second-order Hessian of the loss with respect to model parameters can be computed and interpreted. Importantly, the proof of Theorem 4.2 does not rely on these parameter-wise derivations. We provide a separate complete proof in Appendix E, and it is self-contained.
>
> ---
>
> **Q3: on the scale of experiments**
>
> > The experiments are not rigorous enough... In addition, the reason behind the choice of QKNorm for comparison is unclear.
>
> > Q3: As there are a lot normalization schemes reviewed in Section 2, why do you select QKNorm for comparison?
>
> **Response:** QKNorm is a widely adopted normalization technique in modern large language models and has been used in frontier models such as DeepSeek, Qwen, Kimi, and MiniMax. A useful overview can be found in Sebastian Raschka’s *LLM Architecture Gallery*. Since QKNorm is already common in mainstream architectures, we chose it as one of our primary baselines.
>
> Following your suggestion, we additionally included SandwichNorm as another comparison. We conducted experiments on LLaMA2-7B, and our method still performs clearly better than SandwichNorm. The corresponding results are provided in Figure 1 at the anonymous link below. We can see that SandwichNorm cannot enable large learning rate, and SimpleGPT-7B outperforms LLaMA2-7B with SandwichNorm.
>
> ---
>
> **Q4: on discussion of the limitations of this work**
>
> > The authors have not discussed the limitations of their work. The experiment has room for improvement, as noted in the third weakness mentioned above.
>
> **Response:** As discussed in the paper, our method introduces an additional normalization operation after the linear projection, which incurs some extra computation. More specifically, this overhead is primarily **memory-IO-bound** rather than **compute-bound**: the additional FLOPs are small, but the operation requires extra memory access. In practice, this cost can be reduced significantly through operator fusion using Triton or CUDA. Empirically, as reported in the paper, the end-to-end computational overhead of SimpleGPT is only about 3%. We will make this explanation clearer in the revised version.
>
> ---
>
> _We really hope you are satisfied with our responses._
>
> [1] [Anonymous link](https://shorturl.at/AjYHS)

---

> > ### Author Rebuttal · Reviewer_iFri · 2026-04-03
> >
> > Thank you very much for your detailed response. It clarifies my concerns, especially with regard to the first point. I will update my score accordingly.

---

> > > ### Author Response · Authors · 2026-04-03
> > >
> > > Thank you very much for your positive feedback. We are glad that our clarification, especially on the first point, addressed your concerns.
> > >
> > > We really appreciate your time and consideration, and would be grateful if you could update the score when convenient.

---

### Official Review · Reviewer_fWgK · 2026-03-12

**Soundness:** 3
**Presentation:** 3
**Significance:** 3
**Originality:** 3
**Overall Recommendation:** 5
**Confidence:** 3

**Summary:**

This paper proposes SimpleNorm, a normalization strategy that inserts RMSNorm directly after every linear projection in a Transformer, effectively treating “linear layer + normalization” as a single operator. Based on this, the authors introduce SimpleGPT, which replaces all linear projections (Q, K, V, output, and MLP projections) with the SimpleNorm operator and removes prenorm.

The paper argues that this placement stabilizes activation scale and smooths curvature, enabling larger learning rates and more stable optimization. Empirically, the method is evaluated on nanoGPT, LLaMA2 (1B, 7B), and LLaMA3 (8B), showing consistent reductions in training and validation loss compared to standard prenorm and QKNorm baselines.

**Compliance With Llm Reviewing Policy:**

Affirmed.

**Final Justification:**

My main concerns were about downstream evaluation and missing ablations; both were addressed well in the rebuttal, which strengthens the paper.

**Key Questions For Authors:**

1. Do the gains in training loss translate into improvements on standard evaluation benchmarks (e.g., perplexity on held-out sets, downstream tasks, zero-shot evaluation)?

2.  Is normalization after every linear layer necessary? Have you tested partial insertion strategies?

3. Have you measured curvature (e.g., empirical Hessian spectral norm or gradient variance) to support the smoothness claim experimentally?

These would strengthen the paper.

**Limitations:**

Not adequately. The evaluation focuses primarily on training loss rather than broader performance metrics, coupled with limited analysis of inference cost or deployment implications. A more explicit discussion of these points would strengthen the work.

**Strengths And Weaknesses:**

Strengths

1.	The core proposal is conceptually simple and easy to implement. The structural difference between GPT and SimpleGPT is clearly illustrated.

2.	The related work section is strong. The paper clearly distinguishes its contribution from other normalization strategies. The distinctions are precise and help situate the work properly within existing normalization literature. The mathematical section is also well written and easy to follow.

3.	The method is tested across nanoGPT, LLaMA2, and LLaMA3 at multiple scales (1B, 7B, 8B). The gains appear consistent across architectures, which strengthens the claim that this is a generally applicable modification.

4. The largest-tolerable-learning-rate experiments are useful and directly test the optimization stability claim. It is convincing that SimpleNorm allows higher learning rates compared to prenorm.

5. The paper provides a concrete explanation for why this might help: (i) activation norms are controlled to scale with √d, and (ii) curvature is bounded via Hessian spectral norm arguments. While not fully comprehensive, this gives a more principled justification than purely empirical normalization tweaks.


Weaknesses


1. The improvements are primarily shown in training and validation loss curves. It would be helpful to see whether these gains translate into meaningful downstream performance improvements (e.g., evaluation benchmarks or zero-shot performance), rather than just slightly lower loss.

2. The method inserts normalization after every linear layer. However, there is no ablation isolating whether all placements are necessary. For example:
	•	What happens if normalization is applied only to QKV?
	•	Only to MLP layers?
	•	Only to attention output projections?
Without such ablations, it is unclear whether the full insertion strategy is required.

3. The Hessian-based smoothness argument is interesting, but it is local and activation-level. There is no empirical measurement of curvature, Hessian spectra, or gradient to validate the theory in practice.

Minor Comments:

1.	There are formatting inconsistencies in lines 434–435 regarding capitalization and naming of nanoGPT, LLaMA2, and LLaMA3.

---

> ### Author Rebuttal · Authors · 2026-03-31
>
> We sincerely thank the reviewer for appreciating the contributions of this paper. _We believe understanding of Transformer models from the perspective of second-order or higher-order geometry deserves more attention._
>
> We hope our responses address your concerns.
>
> **Q1: on the evaluation methods**
>
> > The improvements are primarily shown in training and validation loss curves. ,..., into meaningful downstream performance improvements (e.g., evaluation benchmarks or zero-shot performance), ....
>
> **Response:** Thank you very much for your comment. We compare SimpleGPT-7B with LLaMA2-7B and LLaMA2-7B with QKNorm on the C4 validation set, Wiki, Winogrande, PIQA and Helloswag. The results are shown in Table 1. SimpleGPT-7B achieves a better bpb on C4 val and Wiki. It also performs better on Helloswag and Winogrande, and achieves similar performance to LLaMA2-7B with QKNorm on PIQA.
>
> **Table 1.** Comparison of SimpleGPT-7B, LLaMA2-7B (QKNorm), and LLaMA2-7B.
> |Dataset|SimpleGPT-7B|LLaMA2-7B (QKNorm)|LLaMA2-7B|
> |:---:|:---:|:---:|:---:|
> |C4 val（bpb (bits-per-byte)）&darr;|0.7999|0.8327|0.8371|
> |Wiki.（bpb）&darr; |0.7896| 0.8188|0.8227|
> |Winogrande（Acc）&uarr;|53.99|52.88|49.65|
> |PIQA（Acc_norm）&uarr;|65.61|65.77|64.03|
> |Helloswag（Acc_norm &uarr;|45.92|45.33|43.40|
>
> ---
> **Q2: on the ablation of where to insert normalization**
>
> > The method inserts normalization after every linear layer. However, there is no ablation isolating whether all placements are necessary. For example: ..., Without such ablations, it is unclear whether the full insertion strategy is required.
> > Is normalization after every linear layer necessary? Have you tested partial insertion strategies?
>
> **Response:** According to your suggestion, we conducted more ablation studies. Using LLaMA2-7B as our base, we ablate adding SimpleNorm to only attention (Q-K-V-O) and only SwiGLU (W1-W2-W3). The results are shown in Figure 2 of the anonymous link at the end.
>
> We make the following observations:
> - Only adding SimpleNorm to FFN is worse than that only adding  SimpleNorm to self-attention.
> - The full SimpleGPT architecture achieves better and more stable performance than either exclusive counterpart.
>
> ---
> **Q3: on the empirical measurement of curvature**
>
> > The Hessian-based smoothness argument is interesting, but it is local and activation-level. There is no empirical measurement of curvature, Hessian spectra, or gradient to validate the theory in practice.
>
> > Have you measured curvature, ..., to support the smoothness claim experimentally? These would strengthen the paper.
>
> **Response:** Given a linear projection $\mathbf{y} = \mathbf{W} \mathbf{x},$ if $\mathbf{H}\_{\mathbf{y}\mathbf{y}}$ is the Hessian matrix with respect to $\mathbf{y}$, then we can compute the hessian with respect to $\mathbf{x}$ as $$\mathbf{H}\_{\mathbf{x}\mathbf{x}} = (\mathbf{J}\_{\mathbf{x}}^{\mathbf{y}})^\top
> \mathbf{H}\_{\mathbf{y}\mathbf{y}}
> \mathbf{J}\_{\mathbf{x}}^{\mathbf{y}} =
> \mathbf{W}^{\top} \mathbf{H}\_{\mathbf{y}\mathbf{y}} \mathbf{W}.$$
>
> Hence, if the spectral norm of $\mathbf{J}\_{\mathbf{x}}^{\mathbf{y}}$ is large, then $\mathbf{H}\_{\mathbf{x}\mathbf{x}}$ tends to be large.
>
> SimpleNorm using RMSNorm is defined as $$\mathbf{y} = \text{RMSNorm}(\mathbf{W} \mathbf{x}).$$
>
> Given the Hessian matrix $\mathbf{H}\_{\mathbf{y} \mathbf{y}}$,
> its Hessian matrix of $\ell$ with respect to $\mathbf{x}$ is $\mathbf{H}\_{\mathbf{x}\mathbf{x}}=(\mathbf{J}\_{\mathbf{x}}^{\mathbf{y}})^\top
> \mathbf{H}\_{\mathbf{y}\mathbf{y}}
> \mathbf{J}\_{\mathbf{x}}^{\mathbf{y}}+\mathbf{C} = \mathbf{L} + \mathbf{C}$, where $\mathbf{L}$ is the linear term and $\mathbf{C}$ is the curvature term. Similarly, if $\mathbf{J}\_{\mathbf{x}}^{\mathbf{y}}$ is large, then $\mathbf{H}\_{\mathbf{x}\mathbf{x}}$ tends to be large. We can compute $\mathbf{J}\_{\mathbf{x}}^{\mathbf{y}}$ to partly measure curvature.
>
> We record statistics for both the SimpleGPT and LLaMA2 model and find that SimpleGPT has a smaller $\mathbf{J}_{\mathbf{x}}^{\mathbf{y}}$ than LLaMA2. Consequently, SimpleGPT should have a smoother landscape.
>
> ---
> **Q4: on the writing**
>
> > There are formatting inconsistencies in lines 434–435 regarding capitalization and naming of nanoGPT, LLaMA2, and LLaMA3.
>
> **Response:** Thank you for your careful reading. We will correct these issues in the revised version and make the naming conventions consistent throughout the paper.
>
> ---
> **Q5: on more discussion**
>
> > Not adequately. The evaluation focuses primarily on training loss rather than ... A more explicit discussion of these points would strengthen the work.
>
> **Response:** We supply more results in Table 1. The overall computational overhead in the training stage of SimpleGPT is around 3%. SimpleNorm also incurs a slight increase in inference cost. In practice, this cost can be largely reduced through operator fusion using Triton or CUDA.
>
> ___
> [1] [Anonymous link](https://shorturl.at/AjYHS)

---

> > ### Author Rebuttal · Reviewer_fWgK · 2026-04-03
> >
> > Thank you for the detailed and constructive rebuttal. I appreciate the additional experiments and clarifications. Hence, I have increased my score.

---

### Official Review · Reviewer_QetN · 2026-03-12

**Soundness:** 3
**Presentation:** 3
**Significance:** 3
**Originality:** 3
**Overall Recommendation:** 5
**Confidence:** 3

**Summary:**

The paper introduces SimpleGPT, a novel Transformer architecture that stabilizes optimization by placing a normalization layer termed SimpleNorm immediately after every linear projection. The authors provide a theoretical analysis based on second-order geometry, demonstrating that this placement stabilizes intermediate activation scales and decouples the Hessian's spectral norm from the growing spectral norm of the weight matrices. This smoothed loss landscape allows for learning rates 3x to 10x larger than standard configurations. Empirical results across models ranging from 1B to 8B parameters show improved training stability and lower training losses compared to baselines like QKNorm.

**Compliance With Llm Reviewing Policy:**

Affirmed.

**Final Justification:**

The authors clarified my questions, and I have risen my score consequently.

**Key Questions For Authors:**

- How does the addition of SimpleNorm after every linear layer impact the memory footprint during training, specifically regarding activation memory required for backpropagation?
- The proof for Theorem 4.1 assumes weight matrices have a high effective rank ($||W||_F^2 /||W||_2^2\ge cd$). Are your stability benefits (like scale-invariance) impacted during early training phases if weight matrices have not yet reached this high effective rank?
- Could you elaborate on the empirical hypothesis that SimpleNorm increases the depth of nonlinear interactions?
- Are there experiments that can separate the model's improved learning ability from its improved training stability?

**Limitations:**

yes

**Strengths And Weaknesses:**

**Strengths:**
- The proposed architecture and its link to optimization is rigorously analyzed
- It allows for larger learning rates and faster, stable training

**Weaknesses:**
- The assumptions for the benefits of the proposed architecture may be unrealistic (hight effective rank and non-pathological gradient alignment)
- The paper analyzes the idea entirely on training and validation loss on one dataset. There’s no verification on downstream tasks. It’d be interesting to verify if the improved loss translates to better language understanding or even reasoning capabilities.

---

> ### Author Rebuttal · Authors · 2026-03-31
>
> We sincerely thank the reviewer for appreciating our paper. _In this paper, we aim to provide a deeper theoretical understanding of Transformer models from the lens of second-order geometry._
>
> We hope the responses below address your concerns.
>
> **Q1: on the assumptions of our method**
>
> > *- The assumptions for the benefits of the proposed architecture may be unrealistic (high effective rank and non-pathological gradient alignment)*
>
> **Response:** Thank you very much for your comment. For a linear projection, $\mathbf{y} = \mathbf{W} \mathbf{x}$, if $\mathbf{H}\_{\mathbf{y}\mathbf{y}}$ is the Hessian matrix with respect to $\mathbf y$, then by the chain rule $$\mathbf{H}_{\mathbf{x}\mathbf{x}} = \mathbf{W}^{\top}\mathbf{H}\_{\mathbf{y}\mathbf{y}}\mathbf{W}.$$
>
> Hence, $\operatorname{Rank}(\mathbf{H}_{\mathbf x \mathbf x}) \le \operatorname{Rank}(\mathbf{W})$. Therefore, if $\mathbf{W}$ is extremely low-rank, then $\mathbf{H}\_{\mathbf{x}\mathbf{x}}$ is also highly rank-deficient, meaning there are many flat directions and the local curvature is degenerate. Thus, the assumption that $\mathbf{W}$ is not low-rank is reasonable.
>
> When we mention "high effective rank and non-pathological gradient
> alignment," we mean neither $\mathbf{W}$ nor $\mathbf{W}^{\top} \mathbf{H}\_{\mathbf{y}\mathbf{y}} \mathbf{W}$ are low-rank. Otherwise, training will collapse.
>
> ---
> **Q2: on the experiments**
>
> > *- The paper analyzes the idea entirely on training and validation loss on one dataset. There’s no verification on downstream tasks. It’d be interesting to verify if the improved loss translates to better language understanding or even reasoning capabilities.*
>
> **Response:** We provide additional evaluation results in Table 1 (please refer to Table 1 in our response to **Reviewer fWgK**). Specifically, we compare SimpleGPT with LLaMA2-7B and LLaMA2-7B with QKNorm on several downstream tasks. The results show that our method achieves better results than the baselines on several tasks according to bpb and accuracy.
>
> ---
> **Q3: on the influence of SimpleNorm after every linear layer**
>
> > *-   How does the addition of SimpleNorm after every linear layer impact the memory footprint during training, specifically regarding activation memory required for backpropagation?*
>
> **Response:** We report the memory consumption below. We run LLaMA2-7B on 8 A800 GPUs, and our global training size is $8\times 6 \times 4 \times 2048$, where 8 is GPUs count, 6 is gradient accumulation steps, 4 is local batch size per GPU, and 2048 is sequence length.
>
> The memory consumption is as follows: LLaMA2-7B with PreNorm: 54.4 GB; LLaMA2-7B with PreNorm+QKNorm: 58.5 GB; SimpleGPT: 60.5 GB.
>
> Adding several normalizations moderately increases memory consumption during training. We use `torch.compile`, which can fuse some operators and reduce intermediate memory usage, so the observed overhead remains manageable. Further kernel fusion with Triton or CUDA could reduce this overhead.
>
> ---
> **Q4: on the depth of nonlinear interaction**
>
> > *-   Could you elaborate on the empirical hypothesis that SimpleNorm increases the depth of nonlinear interactions?*
>
> **Response:** We are happy to clarify this point. In the paper, we present the SimpleNorm module as
> $$\mathbf{y} = \text{RMSNorm}(\mathbf{W} \mathbf{x}).$$
>
> The Hessian of $\ell$ with respect to $\mathbf{x}$ is
> $$\mathbf{H}\_{\mathbf{x}\mathbf{x}}=\nabla^2_{\mathbf{x}} \ell=
> (\mathbf{J}\_{\mathbf{x}}^{\mathbf{y}})^\top
> \mathbf{H}\_{\mathbf{y}\mathbf{y}}
> \mathbf{J}\_{\mathbf{x}}^{\mathbf{y}}
> +
> \mathbf{C} = \mathbf{L} + \mathbf{C}.
> $$
>
> where $\mathbf{L}$ is the Gauss--Newton (or linearized) term and $\mathbf{C}$ is the nonlinear term induced by the curvature of the normalization.
>
> Although we claim that $\mathbf{C}$ is typically smaller than $\mathbf{L}$, it is nonzero and accumulates across layers. In this sense, each SimpleNorm layer adds one source of nonlinearity to the network, making the overall interaction between layers deeper than in a purely linear case.
>
> ---
> **Q5: on the separation of learning ability and training stability**
>
> > *- Are there experiments that can separate the model's improved learning ability from its improved training stability?*
>
> **Response:** We agree that disentangling improved learning ability from improved training stability is important, although a clean decoupling is inherently challenging. In general, when the learning rate is sufficiently small, training stability is much less likely to be the dominant concern. Based on this consideration, we compared PreNorm, PreNorm with QKNorm, and SimpleNorm under multiple small-learning-rate settings. The results, shown in Figures 5, 8, and 10 in our submission, indicate that even in regimes where training stability is not a major issue, SimpleNorm still consistently outperforms the other two baselines. These results provide evidence that SimpleNorm also offers stronger learning ability.
>
> ---
> [1] [Anonymous link](https://shorturl.at/AjYHS)

---

> > ### Author Rebuttal · Reviewer_QetN · 2026-04-01
> >
> > Thank you for your answers who clarified my questions. I'll raise my score consequently.

---

### Decision · Program_Chairs · 2026-04-30

**Decision:**

Accept (regular)

**Comment:**

All reviewers agree that the paper is well motivated and backed by strong experiments. Regarding the reviewer consensus I recommend acceptation of the paper.